# A phospho-switch controls RNF43-mediated degradation of Wnt receptors to suppress tumorigenesis

Tadasuke Tsukiyama [1✉], Juqi Zou[2], Jihoon Kim [3,4], Shohei Ogamino [5], Yuki Shino[6], Takamasa Masuda[7], Alessandra Merenda[3], Masaki Matsumoto [8,9], Yoichiro Fujioka[10], Tomonori Hirose [11], Sayuri Terai[1], Hidehisa Takahashi[1,11], Tohru Ishitani [2,5,7], Keiichi I. Nakayama [8,12], Yusuke Ohba [10], Bon-Kyoung Koo [4✉] & Shigetsugu Hatakeyama[1]

Frequent mutation of the tumour suppressor RNF43 is observed in many cancers, particularly colon malignancies. RNF43, an E3 ubiquitin ligase, negatively regulates Wnt signalling by inducing degradation of the Wnt receptor Frizzled. In this study, we discover that RNF43 activity requires phosphorylation at a triplet of conserved serines. This phospho-regulation of RNF43 is required for zebrafish development and growth of mouse intestinal organoids. Cancer-associated mutations that abrogate RNF43 phosphorylation cooperate with active Ras to promote tumorigenesis by abolishing the inhibitory function of RNF43 in Wnt signalling while maintaining its inhibitory function in p53 signalling. Our data suggest that RNF43 mutations cooperate with KRAS mutations to promote multi-step tumorigenesis via the Wnt-Ras-p53 axis in human colon cancers. Lastly, phosphomimetic substitutions of the serine trio restored the tumour suppressive activity of extracellular oncogenic mutants. Therefore, harnessing phospho-regulation of RNF43 might be a potential therapeutic strategy for tumours with RNF43 mutations.

[1] Department of Biochemistry, Faculty of Medicine and Graduate School of Medicine, Hokkaido University, Kita 15, Nishi 7, Kita-ku, Sapporo, Hokkaido 060-8638, Japan. [2] Department of Homeostatic Regulation, Research Institute for Microbial Diseases, Osaka University, 3-1 Yamadaoka, Suita, Osaka 565-0871, Japan. [3] Department of Genetics, University of Cambridge, Cambridge CB2 3EH, UK. [4] Institute of Molecular Biotechnology of the Austrian Academy of Sciences (IMBA), Vienna Biocenter (VBC), Dr. Bohr-Gasse 3, 1030 Vienna, Austria. [5] Laboratory of Integrated Signaling Systems, Department of Molecular Medicine, Institute for Molecular and Cellular Regulation, Gunma University, 3-39-15 Showa-machi, Maebashi, Gunma 371-8512, Japan. [6] Faculty of Medicine, Hokkaido University, Kita 15, Nishi 7, Kita-ku, Sapporo, Hokkaido 060-8638, Japan. [7] Division of Cell Regulation Systems, Medical Institute of Bioregulation, Kyushu University, 3-1-1 Maidashi, Higashi-ku, Fukuoka 812-8582, Japan. [8] Division of Proteomics, Research Center for Transomics Medicine, Medical Institute of Bioregulation, Kyushu University, 3-1-1 Maidashi, Higashi-ku, Fukuoka 812-8582, Japan. [9] Department of Omics and Systems Biology, Niigata University Graduate School of Medical and Dental Sciences, 1-757 Asahimachi-douri, Chuo-ku, Niigata, Niigata 951-8510, Japan. [10] Department of Cell Physiology, Faculty of Medicine and Graduate School of Medicine, Hokkaido University, Kita 15, Nishi 7, Kita-ku, Sapporo, Hokkaido 060-8638, Japan. [11] Department of Molecular Biology, Yokohama City University Graduate School of Medical Science, Fukuura 3-9, Kanazawa-ku, Yokohama, Kanagawa 236-0004, Japan. [12] Division of Cell Biology, Department of Molecular and Cellular Biology, Medical Institute of Bioregulation, Kyushu University, 3-1-1 Maidashi, Higashi-ku, Fukuoka 812-8582, Japan. ✉email: tsukit@med.hokudai.ac.jp; bonkyoung.koo@imba.oeaw.ac.at

Tight regulation of the many signalling pathways that control cell fate is essential for proper development and homoeostasis. The Wnt signalling pathway plays a prominent role in stem cell maintenance[1–3], embryonic development[4] and tumorigenesis[5–8]. The activities of key players in this pathway are regulated by post-translational modifications, such as phosphorylation and ubiquitination[9,10]. Loss of these regulatory events often induces oncogenic transformation.

The intestinal stem cell (ISC)-specific ubiquitin ligase ring finger protein 43 (RNF43) negatively regulates Wnt signalling by triggering ubiquitin-mediated, endo-lysosomal degradation of the Frizzled (Fzd) family of Wnt receptors[11]. RNF43-mediated negative regulation of Fzd requires an interaction between the extracellular domains of RNF43 and Fzd. The growth factor R-spondin (Rspo) is a vertebrate-specific Wnt agonist that reversibly counteracts negative regulation by RNF43 and also by its homologue zinc and ring finger protein 3 (ZNRF3)[12]. Rspo binds the stem cell-specific leucine-rich repeat-containing G-protein-coupled receptors Lgr4/5/6 and promotes formation of a trimeric complex with RNF43/ZNRF3[13]. As a consequence, RNF43/ZNRF3 target Lgr4/5/6 for degradation instead of Fzd, leading to an accumulation of Fzd Wnt receptors and subsequent elevation of Wnt signalling activity[13,14]. Thus, Wnt signalling undergoes complex regulation not only in the cytoplasm by the β-catenin destruction complex[8] but also at the plasma membrane by Rspo and RNF43/ZNRF3.

RNF43 itself is a downstream target gene of Wnt/β-catenin signalling[15] and thus acts as an important negative feedback regulator to inhibit an excess of Wnt signal activity. Important hub proteins of Wnt signalling are frequently mutated in various types of cancer[16]. Indeed, missense or truncation mutations of RNF43 that compromise the negative feedback regulation of Wnt signalling have been identified in cancer[17,18]. Tumours with these mutations are still dependent on Wnt for growth, but independent of Rspo[18]. Missense mutations of RNF43 and ZNRF3 appear to function in a dominant-negative manner[18], consistent with the observation that many cancers exhibit a single mutation in either RNF43 or in ZNRF3 but not both. Many RNF43 cancer-associated mutations occur outside the N-terminal extracellular and cytoplasmic RING-finger domains, which regulate Fzd binding and ubiquitination, respectively, suggesting additional levels of RNF43 regulation. We previously showed that the C-terminal cytoplasmic region of RNF43 interacts with the Wnt signal transducer Dishevelled (Dvl) to suppress non-canonical Wnt signalling[18]. This work also revealed that amino acid residues 442-478 within the cytoplasmic region of RNF43, which are not involved in the interaction with Dvl[18,19], are important for RNF43 regulation.

In addition, RNF43 suppresses p53-dependent transcription and cell death that are induced by DNA damage[20] or viral infection[21]. However, the molecular significance of oncogenic RNF43 mutations in the p53 pathway remained an unsolved issue.

Here, we discover that intracellular phosphorylation of RNF43 is required for negative regulation of Wnt signalling. RNF43 mutations that abrogate this phosphorylation lead to loss of Fzd ubiquitination-degradation and de-repression of Wnt signalling, but do not compromise inhibition of p53. These RNF43 phospho-mutants cooperate with active Ras to promote tumorigenesis, and co-occurrence of RNF43 and KRAS mutations are associated with poor survival in human colon cancer[22]. Strikingly, introducing phosphomimetic mutations into oncogenic RNF43 mutants restores RNF43-mediated ubiquitination and degradation of Fzd and inhibition of Wnt signalling, and abolishes oncogenic RNF43-Ras-mediated tumorigenesis in vitro and in vivo. Our results reveal RNF43 as a potential therapeutic target, and provide important insights into the mechanisms that promote multi-step colon cancer tumorigenesis along the Wnt-Ras-p53 axis.

## Results

**Multi-step phosphorylation of serines activates RNF43.** Our previous work suggested that RNF43 binding to Dvl2 is not required to downregulate Fzd4[18]. To verify that Dvl is not an essential cofactor for the recognition and degradation of Fzd, we determined whether the interaction with Dvl is required to regulate Fzd5. We found that a Dvl-interaction-defective RNF43 mutant (RNF43-ΔDvl-C) can still immunoprecipitate Fzd5 (Supplementary Fig. 1a, b) and induce its downregulation to the same degree as Dvl-interacting RNF43 in STF-Luc Wnt reporter assays using STF293 cells (i.e., HEK293 cells genetically carrying the SuperTopFlash-luciferase (STF-Luc) Wnt reporter, which is a specific and sensitive detector of endogenous Wnt activity) with Wnt3a conditioned media (CM) and flow cytometry (Supplementary Fig. 1c, d). These data further suggest that the RNF43-Dvl interaction is dispensable for RNF43-mediated regulation of surface Fzd expression and Wnt/β-catenin signalling in this context.

We have reported that RNF43 amino acids 442–478, close to known Dvl binding regions, are important for RNF43 regulation[18,19]. We turned our attention to two serine-rich regions (SRR-1: aa 442–449, SRR-2: aa 466–478) that are similar to a region within β-catenin. SRR-2 in RNF43 displays partial conservation with its homologue ZNRF3, while SRR-1 does not (Supplementary Fig. 1e). STF-Luc assays with RNF43 constructs containing deletions and point mutations in these regions indicated that three consecutive serine residues in SRR-2-2 (containing a serine triplet; S474, S475, S476), which are structurally and functionally conserved in ZNRF3 and RNF43 in other species, were indispensable for RNF43-mediated suppression of Wnt signalling (Fig. 1a–c, Supplementary Fig. 1e, f, h). Based on these facts, we hypothesised that these serines are regulated by phosphorylation. To test this hypothesis, we replaced all three serines with aspartic (3SD) or glutamic acid (3SE) residues to mimic the phosphorylated form of serine. We found that the phosphomimetic RNF43 mutant maintained negative regulation of Fzd and Wnt/β-catenin signalling (Fig. 1c, d, Supplementary Fig. 1g). Conversely, replacing these serines in RNF43 with alanine (3SA) to prevent phosphorylation, or with threonine (3ST), impaired Fzd regulation and Wnt signalling compared to wild-type (WT) RNF43, despite showing similar expression levels to WT RNF43 (Fig. 1c, d, Supplementary Fig. 1g). However, RNF43(3SA) phospho-mutant did not autonomously activate Wnt signalling but facilitated the signal activity in a Wnt ligand-dependent manner (Supplementary Fig. 1i). Together, these results suggest that phosphorylation of the serine triplet is necessary for RNF43 function.

To demonstrate that these serines are phosphorylated in the normal cellular environment, we performed Phos-tag SDS-PAGE, which can distinguish phosphorylated proteins by a band-shift. This analysis indicated that a signal corresponding to phospho-RNF43 was lost in the phospho-deficient mutant RNF43(3SA) (Supplementary Fig. 2a). In addition, 2D-PAGE revealed the loss of a phospho-RNF43 signal in cells expressing RNF43(3SA), or when cells expressing RNF43(WT) were cultured under phosphate-depleted conditions (Supplementary Fig. 2b, c). Furthermore, $^{32}$Pi metabolic labelling of cells revealed significantly lower phosphorylation of RNF43(3SA) than RNF43(WT) (Fig. 1e, Supplementary Fig. 2d). These data support our hypothesis that under normal cellular conditions RNF43 phosphorylation regulates its function.

Our previous report suggested that S478 is also indispensable for regulating RNF43 function[18]. Indeed, we found that substitution of S478 with A or D/E led to the inhibition or activation, respectively, of RNF43 function in STF-Luc assays, similar to substitutions at the conserved serine triplet (Fig. 2a).

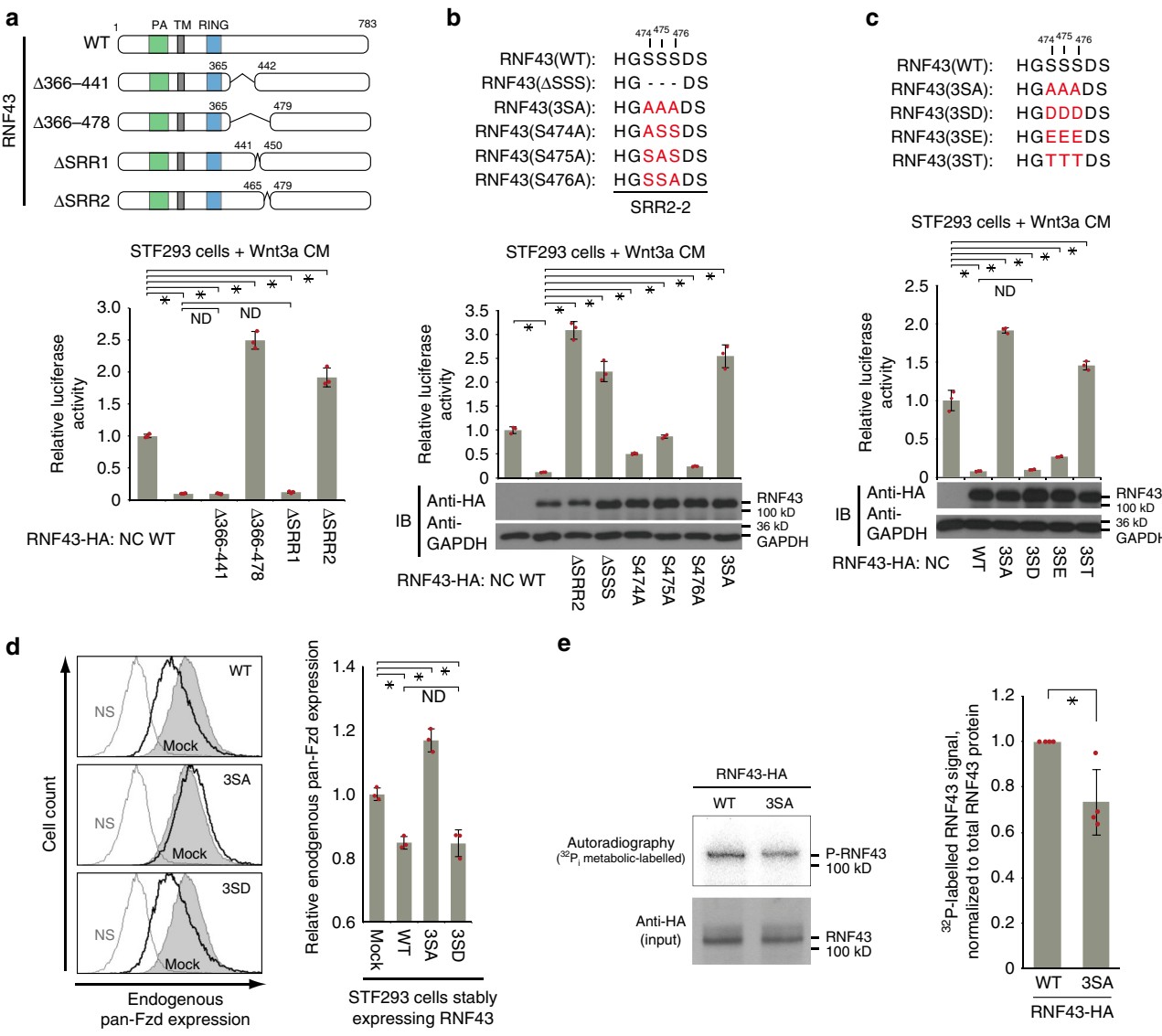

**Fig. 1 RNF43 function requires phosphorylation of a serine triplet. a–c** Regulation of Wnt3a-mediated activation of Wnt/β-catenin signalling by RNF43 mutants was examined using SuperTopFlash (STF)-luciferase reporter assays. Luciferase activity in empty vector-transfected negative control (NC) cells was set to 1. SRR: serine rich region. Characters shown in red indicate amino acids after substitution (**b, c**). PA: protease-associated domain. TM: transmembrane region. RING: RING finger domain. **d** Expression of frizzled (Fzd) at the surface of RNF43 mutant-expressing cells was measured via flow cytometric analysis. FACS data was acquired and displayed with same strategy shown in Supplementary Fig. 1g. Grey or black lines, or grey fills indicate not stained, RNF43 stably expressed or mock cells, respectively. Expression in mock-transfected cells was set to 1. **e** Cellular phosphorylation of RNF43 was examined using $^{32}P_i$ metabolic labelling. Radio-labelled RNF43 levels were normalised to total RNF43 protein levels. The relative phospho-RNF43 level in mock-transfected cells was set to 1. Bar graphs and error bars in this figure represent mean ± standard deviation (sd) of at least three biologically independent experiments. Red circles indicate individual values of each sample. The *P* values for the indicated comparisons were determined by one-way analysis of variance (ANOVA) ($P < 0.05$). $n = 3$ (**a–d**), $n = 4$ (**e**) biologically independent samples. Asterisks or ND indicates significant or no significant difference in indicated comparisons, respectively.

The S478A substitution in RNF43 increased the cell surface levels of Fzd compared to WT RNF43 (Fig. 2b), suggesting that the S478 phospho-status also regulates Fzd levels and Wnt signalling.

The serine-rich region of β-catenin, which is similar to SRR2 of RNF43[23,24], is sequentially phosphorylated by CK1 and GSK-3β[25]. Therefore, we examined whether stepwise phosphorylation also regulates RNF43. Our STF-Luc reporter assay showed that RNF43 containing 3SA-S478D substitutions impaired its function, whereas RNF43(3SD-S478A) retained function (Fig. 2c). These data suggest that S478 phosphorylation might be upstream of serine triplet phosphorylation and required only to prime the subsequent phosphorylation of the serine triplet.

Next, we examined whether CK1 or GSK-3β phosphorylate RNF43. Flow cytometric analysis showed that GSK-3β inhibition did not alter RNF43-mediated regulation of Fzd surface levels (Fig. 2d). In contrast, CK1 inhibition led to restored surface levels of Fzd in RNF43(WT)- and RNF43(S478D)-expressing cells, but not in RNF43(3SD)-expressing cells (Fig. 2d). These results suggest that the serine triplet is phosphorylated by CK1. Indeed, an in vitro kinase assay detected the CK1-dependent phosphorylation of RNF43(WT) but not RNF43(3SA) (Fig. 2e, Supplementary Fig. 2e) and also revealed the phosphorylation of endogenous RNF43 by CK1 (Supplementary Fig. 2f). Interestingly, the total amount of phosphorylation signal was significantly

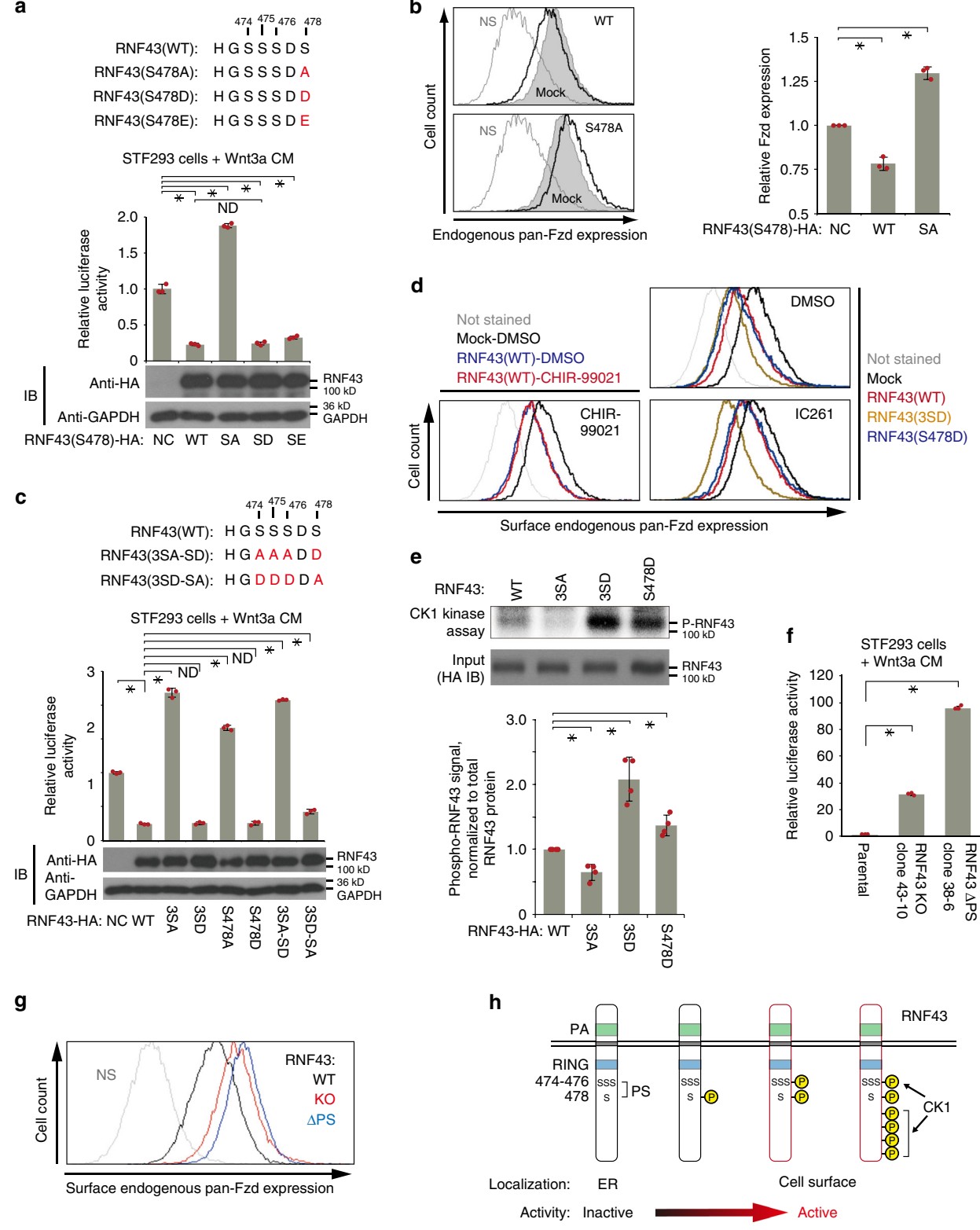

increased with both the RNF43(S478D) and the RNF43(3SD) mutant, suggesting that the phosphorylation of the serine triplet is required for additional RNF43 phosphorylation events outside the SSSDS sequence, which we confirmed by MS/MS analysis (Supplementary Fig. 2g, h). Furthermore, a lower level of phosphorylation was observed on oncogenic RNF43(R127P), which accumulates in the ER (Supplementary Fig. 2j)[18], compared to WT RNF43 (Supplementary Fig. 2i). This result

suggests that RNF43 might be phosphorylated in a localisation-dependent manner and activated after it leaves the ER. Collectively, these data demonstrate that RNF43 activity is regulated by multi-step phosphorylation events (Fig. 2h).

**Exogenous RNF43 functions similarly to endogenous RNF43.** First, we confirmed that the STF293 cells we used in this study

**Fig. 2 Multi-step phosphorylation is required for RNF43 function. a–c** The role of a priming phosphorylation was investigated using STF-Luc assay (**a**, **c**) and flow cytometric analysis (**b**) using RNF43 mutants. Luciferase activity or surface Fzd level in empty vector-transfected (NC) or mock cells was set to 1. Grey or black lines, or grey fills indicate not stained, RNF43 stably expressed or mock cells, respectively. Characters shown in red indicate amino acids after substitution (**a**, **c**). **d** Surface expression of Fzd was examined via flow cytometric analysis following addition of kinase inhibitors (GSK-3β, CHIR-99021; CK1, IC261). **e**, Phosphorylation of RNF43 and mutant forms was examined by an in vitro kinase assay with CK1/2. Phospho-RNF43 levels were normalised to total RNF43 protein levels and normalised phospho-RNF43(WT) levels were set to 1. **f** The effects of a loss of endogenous RNF (KO) or removal of the RNF43 phospho-switch (ΔPS; similar to serine-rich region (SRR)2 and SRR2-2 in Supplementary Fig. 1f) were examined in STF293 cells using a STF-luciferase assay. The luciferase activity in parental STF293 cells was set to 1. **g** Surface Fzd expression on RNF43 KO or ΔPS STF293 cells was evaluated using flow cytometry with pan-Fzd antibodies (Abs). **h** Schematic of localisation-dependent RNF43 activation via multi-step phosphorylation. RNF43 activity is acquired via phosphorylation at a post-ER stage during or after protein trafficking toward the cell surface. Bar graphs and error bars in this figure represent mean ± standard deviation (sd) of at least three biologically independent experiments. Red circles indicate individual values of each sample. The $P$ values for the indicated comparisons were determined by one-way ANOVA ($P < 0.05$). $n = 3$ (**a–c**, **f**), $n = 4$ (**e**) biologically independent samples. Asterisks or ND indicates significant or no significant difference in indicated comparisons, respectively. All FACS data in this figure was acquired and displayed with same strategy shown in Supplementary Fig. 1g. Each coloured line indicates the property of RNF43 expressing in cells (**d**, **g**).

express functional endogenous RNF43 on their cell surface. Rspo facilitates Wnt ligand-induced signal activation by increasing the surface level of Fzd[26,27] (Supplementary Fig. 3a, b). Direct activation of Wnt/β-catenin signalling by CHIR treatment suppressed the expression of Fzd on the surface via a negative feedback circuit, as reported by many groups[15,18] (Supplementary Fig. 3b). Rspo interfered with the suppression of Fzd expression via RNF43 induction (Supplementary Fig. 3b). These results suggested that STF293 cells retain an intact Wnt signal activation/suppression cascade as well as Rspo-Lgr4/5-RNF43/ZNRF3 axis. In contrast to this, CRC cells that express abundant levels of endogenous WT RNF43 due to saturated accumulation of β-catenin by APC mutation (e.g. HT-29 cells), fail to show the ligand-induced signal activation and facilitation (Supplementary Fig. 3a). Therefore, we used STF293 cells and the CRISPR/Cas9 genome engineering method to confirm our findings at the endogenous protein level. We tagged RNF43 with a C-terminal HA-epitope in STF293 cells (Supplementary Fig. 3c, d). Although it still functions normally (Supplementary Fig. 3a, b), endogenous RNF43 protein was expressed at extremely low levels, and thus we could not detect HA-tagged endogenous RNF43 by simple immunoblotting (IB) and immunofluorescent staining (IF) (Supplementary Fig. 3e, g). Nevertheless, we were able to confirm the expression of HA-tagged endogenous RNF43 by IB after immunoprecipitation (IP) (Supplementary Fig. 3e). We confirmed that the HA knock-in STF293 cells show a similar response to WT STF293 cells upon Wnt stimulation (Supplementary Fig. 3f).

Next, we used CRISPR/Cas9 genome editing to introduce various RNF43 mutations (RNF43 KO, ΔPS, and R127P) in STF293 cells and assess their effects at the endogenous level (Supplementary Fig. 3h–k). RNF43 knockout (KO) cells exhibited increased Wnt signalling (Fig. 2f), suggesting that STF293 cells express functional endogenous RNF43 protein as shown in our previous knockdown experiment[18]. Therefore, these cells are useful for a functional examination of endogenous Wnt signalling, despite a low RNF43 expression level. Furthermore, deletion of the phospho-switch region (ΔPS) from endogenous RNF43 (Supplementary Fig. 3j) facilitated Wnt signal activation compared RNF43 KO cells (Fig. 2f). This result confirms our findings from experiments with overexpression of exogenous RNF43 (ΔSRR2, ΔSRR2-2 or 3SA; Fig. 1, Supplementary Fig. 1f) and was accompanied by corresponding changes in the surface levels of endogenous Fzd receptors (Fig. 2g). STF293-RNF43 (ΔPS) and RNF43(R127P) knock-in cells were not sensitive to Rspo (Supplementary Fig. 3l). Taken together, these results suggest similar functions of endogenous and exogenous RNF43, with endogenous RNF43(ΔPS) protein acting as a dominant-negative mutant[18] (Figs. 1, 2 and Supplementary Fig. 1 with exogenous RNF43 mutants).

**RNF43 phosphorylation promotes ubiquitination of Fzd.** To further investigate how RNF43 function is regulated via serine phosphorylation, we examined protein-protein interactions and subcellular localisation of the phospho-mutants. Immunoprecipitation and immunofluorescent experiments suggested that RNF43(3SA) and RNF43(3 SD) behave similarly to RNF43(WT) with regard to protein-protein interactions (Supplementary Fig. 4a), homodimer formation (Supplementary Fig. 4b), heterodimer formation with ZNRF3 (Supplementary Fig. 4c) and endosomal/ER/nuclear localisation (Supplementary Fig. 4d). In addition, expression of *ZNRF3* was not perturbed by any of the RNF43 mutants (Supplementary Fig. 4e).Co-immunoprecipitation analysis also revealed that the RNF43 phosphomimetic (3SD) and phosphoresistant (3SA) mutants maintained the interactions with Dvl2 or Fzd5, similar to WT RNF43 (Supplementary Fig. 4f, g). We then considered that serine phosphorylation may change the conformation of the intracellular portion of RNF43 to expose the RING finger domain, which is essential for interactions with E2 enzymes. However, phospho-mutations did not alter binding to the E2 enzyme UbcH5C, which is essential for RNF43-mediated ubiquitination of Fzd[26] (Supplementary Fig. 4h). Previously, we and other groups reported the localisation of RNF43 in the nuclear membrane[18,28–30] and demonstrated that this protein negatively regulated Wnt/β-catenin signalling by depleting the Tcf4 transcription factor from Wnt target genes[29,30]. Accordingly, we examined the role of the phospho-switch in this mechanism by directly activating nuclear Wnt/β-catenin signalling with ΔN-β-catenin. As reported, WT RNF43 suppressed Wnt/β-catenin signalling downstream of receptor control, although a marginal level of negative regulation was also observed at the nuclear level (Supplementary Fig. 4i). The functions of all other RNF43 phospho-mutants were similar to the WT.

Based on these negative results, we next hypothesised that RNF43 phosphorylation affects its ability to ubiquitinate Fzd. To test this hypothesis, we treated cells co-expressing Fzd5 and RNF43 with bafilomycin to inhibit lysosome-dependent degradation and monitored the levels of polyubiquitinated Fzd5. Indeed, cells co-expressing RNF43(3SA) displayed less polyubiquitinated Fzd5 than cells co-expressing RNF43(3SD) or WT RNF43 (Supplementary Fig. 4j, see also Fig. 6d). These data suggest that phosphorylation promotes RNF43-mediated ubiquitination, and in turn endocytosis and lysosomal degradation, of Fzd5.

It is currently thought that RNF43-mediated degradation of the substrate Fzd is regulated by RNF43 levels and by R-spondin. Our results reveal an additional layer of regulation in Wnt signalling. In general, phosphorylation of the substrate is known to regulate its degradation, however, we have uncovered phospho-regulation of the ubiquitin ligase RNF43 itself as another way to control degradation of Fzd. We clearly show that phosphorylation of

RNF43 directly facilitates its ubiquitin ligase activity. This phospho-dependent regulation may function downstream from the well-known RNF43-Rspo-Lgr module.

**Phospho-RNF43 regulates morphogenesis and ISC maintenance.** We next examined the biological significance of RNF43 phospho-regulation in developing zebrafish embryos. Injection of mRNA encoding RNF43(3SA) into Wnt reporter zebrafish embryos resulted in increased expression of both endogenous Wnt/β-catenin target genes, such as *tbx6*[31], *axin2*[32] and *nkd1*[33,34], as well as the reporter gene, *egfp*. In contrast, injection of mRNA encoding RNF43(WT) inhibited the expression of these genes, as expected (Fig. 3a, b, Supplementary Fig. 5a). The activation and expression of target genes in other signalling pathways, including FGF[35] and BMP[36], were not perturbed by RNF43 expression in developing zebrafish embryos (Fig. 3b). In addition, we observed defective anterior–posterior (A–P) axis elongation at later stages of development in embryos injected with mRNA encoding RNF43(WT), but this defect was absent from embryos injected with mRNA encoding RNF43(3SA) (Supplementary Fig. 5b). Impaired A–P axis elongation can arise from defective mesoderm induction due to a lack of sufficient Wnt/β-catenin activity, or from the loss of convergent extension movement due to deficient non-canonical Wnt signalling. Therefore, we investigated whether non-canonical Wnt signalling activity is altered by RNF43(3SA) expression. In situ hybridisation for the expression of axis-related marker genes showed that the short and wide non-canonical Wnt phenotype[37] induced by expression of RNF43(WT) was mostly absent in embryos expressing RNF43(3SA) (Supplementary Fig. 5c). Together, these data suggest that the defects in A–P axis elongation originate from dysfunction of both canonical Wnt/β-catenin signalling and likely non-canonical Wnt signalling, with minimal influences from other signalling pathways. They also establish a critical role for the S474–476 phospho-switch of RNF43 during embryonic development.

We have reported that RNF43 is expressed in ISCs, where it regulates normal crypt development and stem cell maintenance[11]. RNF43 and ZNRF3 strongly inhibit Wnt/β-catenin signalling to suppress excess proliferation of ISCs, while maintaining Wnt/β-catenin signalling is also important for the self-renewal of ISCs. These paradoxical roles suggest that RNF43 expressed in ISCs might be regulated to fine-tune its activity and maintain ISC self-renewal. Based on this idea, we employed a mouse intestinal organoid model to examine whether the phospho-regulation of RNF43 function contributes to ISC maintenance. Expression of RNF43(3SD) during short-term culture of organoids in the presence of EGF, Noggin and Rspo (the ENR condition) significantly suppressed organoid growth only when the Rspo concentration in the culture medium was low (1% Rspo-conditioned medium). By contrast, expression of RNF43(3SA) did not suppress organoid growth under either low or high (10%) Rspo concentrations but instead improved their viability in the 1% Rspo ENR condition (Fig. 3c, Supplementary Fig. 5d, e). These results suggest that phospho-regulation of RNF43 is required for the survival and the growth of crypts, and that this regulation occurs independently of the RNF43-Rspo-Lgr regulatory module[13]. We then added exogenous Wnt3a to ENR medium (the WENR condition) to rule out a function of the Paneth cell-niche. Again, ISCs failed to maintain the organoid culture after the second passage when expressing RNF43(3SD) (Fig. 3d, Supplementary Fig. 5f), suggesting the importance of phospho-regulation of RNF43 in ISC maintenance during long-term culture. These results reveal that phospho-regulation of RNF43 functions as an additional regulatory layer of Wnt signalling and suggest the functional importance of maintaining

RNF43 in a low-phosphorylation state in ISCs. This RNF43 phospho-regulation ensures tighter control of ISC activity together with the Rspo-Lgr4/5 regulator of RNF43.

**RNF43(3SA) cooperates with active Ras to induce tumours.** Inactivation of RNF43 and ZNRF3 induces Wnt-dependent adenoma formation[11,17,38], and these genes are frequently mutated in various human cancers[39,40]. These studies suggested essential roles of RNF43/ZNRF3 in suppressing tumorigenesis in both mice and humans. Indeed, several cancer-associated mutations in the N-terminal extracellular domain of RNF43 greatly increase Wnt/β-catenin signalling activity in a Wnt-dependent, but Rspo-independent, manner[18]. To further investigate the role of RNF43 in tumorigenesis, we identified patient mutations downstream of the RING-finger domain using the COSMIC database for mutations in cancer. Twenty-three patient tumour mutations within the cytoplasmic region after the RING-finger domain, but outside the SSSDS sequence, did not alter RNF43 function in the STF-Luc reporter assay (Supplementary Fig. 6a), suggesting that these are passenger mutations. In contrast, we found that four naturally occurring cancer-associated mutations within the SSSDS sequence inhibit or are predicted to inhibit RNF43-mediated repression of Wnt/β-catenin signalling (Figs. 1b, 2a, Supplementary Fig. 6b, c), suggesting that loss of RNF43 phospho-regulation contributes to tumorigenesis.

To directly investigate the role of the RNF43 phospho-regulation in tumorigenesis, we established cell lines that stably express RNF43 constructs (Supplementary Fig. 7a). We first used the non-tumour cell line NIH3T3 to avoid unexpected genomic variations that frequently occur in cancer-derived cell lines due to their genomic instability. NIH3T3 cells expressing dominant-negative RNF43(3SA) did not acquire a transformed phenotype, as assessed by anchorage-independent cell growth (Supplementary Fig. 7b, left panel) and allograft transplantation into nude mice (Fig. 4a, left panel). These data suggest that the gain of RNF43-mediated facilitation of Wnt/β-catenin signalling alone is not sufficient for tumorigenesis. Given the high co-occurrence of *RNF43* mutations with activating *KRAS* mutations in human pancreatic tumours[39], we next investigated the oncogenic properties of RNF43(3SA) in NIH3T3 cells that contain mutant, active Ras (Cle-H3 cells)[41]. Strikingly, Cle-H3 cells expressing RNF43(3SA) exhibited greatly accelerated anchorage-independent colony formation, spheroid formation and tumour growth in nude mice (Fig. 4a–d, Supplementary Fig. 7a–c). In fact, RNF43(3SA) displayed similar oncogenic properties as an established cancer-associated R127P mutation in RNF43, which is within the extracellular protease-associated (PA) domain and abolishes inhibition of Wnt signalling[18,29] (Fig. 4a–d, Supplementary Fig. 7c). This suggests that the phospho-switch mutation (3SA) has a similar dominant-negative effect observed in previous known oncogenic mutations found in the PA domain and that the loss of RNF43 phosphorylation cooperates with active Ras to promote tumorigenesis in vivo. These established tumours arising from Cle-H3 cells expressing RNF43(3SA) displayed a strong cytoplasmic accumulation of β-catenin as previously described in *RNF43/ZNRF3* DKO intestine[11] (Supplementary Fig. 7d).

**RNF43-KRAS cooperation triggers multi-step carcinogenesis.** We and others have previously reported that RNF43 inhibits p53-dependent cellular events, including transcription of downstream genes, cell cycle progression and cell death[20,21,42]. To determine whether oncogenic RNF43 mutants maintain inhibition of p53, we treated HCT116 cells (WT p53, active-Ras$^{G13D}$, active-β-catenin$^{S45\Delta}$ and RNF43 deletion) that stably overexpressed exogenous RNF43 and mutants with etoposide and examined the p53

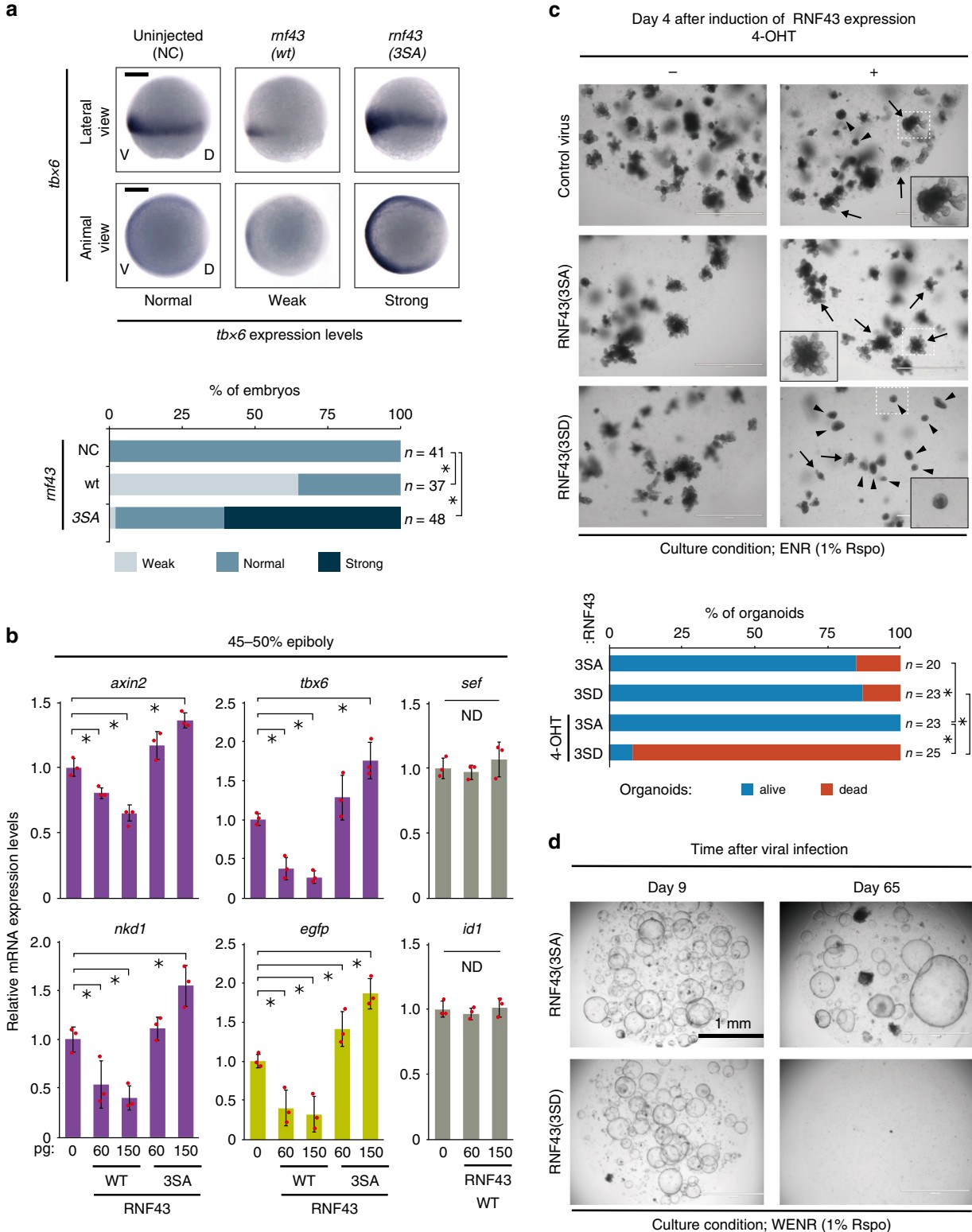

response in the absence of endogenous RNF43 overexpression induced by aberrant Wnt signalling. We found that the RNF43 mutants (3SA, 3SD, R127P) retained DNA damage-induced p53 expression and its nuclear localisation, and suppressed p53-dependent induction of p21 and Bax, similar to WT RNF43, suggesting that the nuclear function of RNF43 does not require phosphorylation and ubiquitinating activity (Fig. 5a, Supplementary Figs. 4i, 8). Furthermore, downregulation of p53 target

genes by RNF43 and its derivatives were completely abolished in MB352 cells with the lack of *TP53* gene, whereas it was maintained in the culture of STF293 cells with iCRT3, which inhibits β-catenin and Tcf/Lef binding (Supplementary Fig. 8a, b). These data provide further support that RNF43 does not degrade p53 and does not suppress p53 target genes via the Wnt-myc-p21 pathway[43] but suppresses p53-dependent transcription[20], and that it controls multiple signalling pathways via distinct

**Fig. 3 RNF43 phosphorylation regulates morphogenesis and ISC maintenance. a** Expression of endogenous target genes of Wnt/β-catenin signalling was evaluated using in situ hybridisation following expression of RNF43 phospho-mutant forms in zebrafish embryos at 8 h post-fertilisation (hpf) (75% epiboly). Scale bars, 200 μm. Asterisks indicate significant differences ($P < 0.05$, one-way ANOVA, $n = 37$–48 as indicated) from NC embryos. **b** Expression change of Wnt/β-catenin, FGF and BMP signalling target genes with RNF43 was evaluated using qPCR at 5.3 hpf (45–50% epiboly). Endogenous Wnt target genes, artificial Wnt reporter gene and non-Wnt target genes are shown in purple, green and grey, respectively. Expression of each gene in uninjected embryos was set to 1 (mean ± sd). Red circles indicate individual values of each sample. Asterisks indicate significant differences ($P < 0.05$, one-way ANOVA, $n = 3$, biological replicates with pools of 25–30 embryos) from uninjected embryos. ND indicates no significant difference. **c** Short-term development of intestinal organoids was examined at 4 days following induction of RNF43 phospho-mutant forms. White dashed boxes denote the area enlarged in black boxes. Arrows, healthy organoids. Arrowheads, dead organoids. Asterisks indicate significant differences ($P < 0.05$, one-way ANOVA, $n = 20$–25 as indicated) between groups. **d** Long-term ISC maintenance was evaluated after 65 days with two passages in organoids expressing RNF43 phospho-mutant forms. Scale bars in (**c**, **d**), 1 mm.

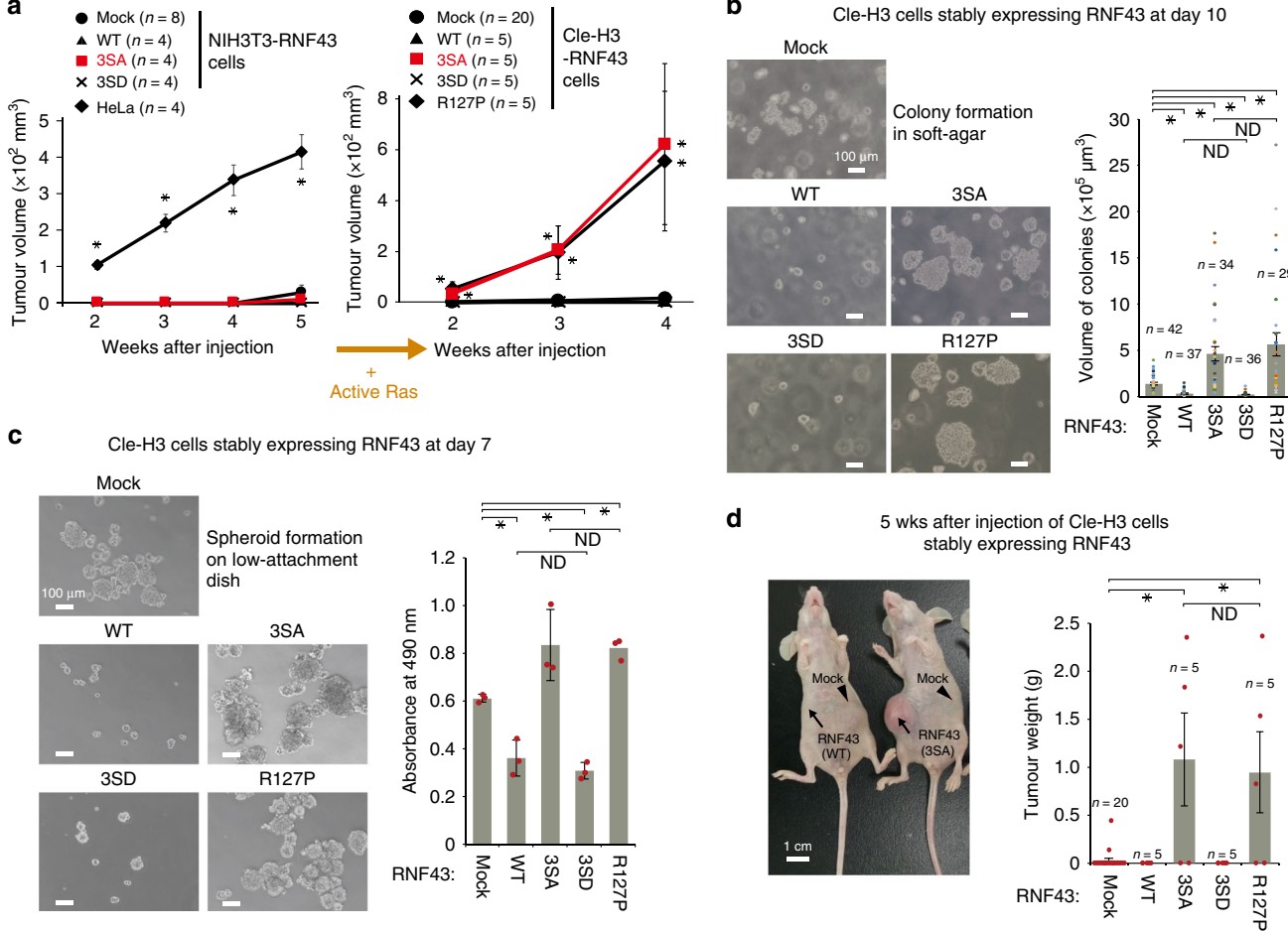

**Fig. 4 Active Ras cooperates with mutant RNF43 to induce tumorigenesis. a** Time-dependent tumour growth was examined in nude mice following expression of RNF43 phospho-mutant forms in NIH3T3 and Cle-H3 cells. Estimated tumour volume is shown. Red lines indicate RNF43(3SA)-expressing tumours. **b** Anchorage-independent colony-forming activity was evaluated via soft agar assay following the expression of RNF43 phospho-mutant forms in Cle-H3 cells. Scale bars, 100 μm. **c** Spheroid-forming activity was examined in Cle-H3 cells expressing RNF43 mutant forms. Spheroid formation was quantified via MTS assay. **d** Image of nude mice injected with Cle-H3-RNF43 cells. Scale bars, 1 cm. Tumour weight was measured at 5 wks following Cle-H3 injection. Graphs and error bars in this figure represent mean ± sd (**c**) or ± standard error of the mean (sem) (**a**, **b**, **d**) of independent experiments or samples. Red circles indicate individual values of each sample **b**–**d**). The $P$ values for the indicated comparisons in this figure were determined by one-way ANOVA ($P < 0.05$). $n = 4$–20 (**a**), $n = 29$–42 (**b**), $n = 3$ (**c**), $n = 5$–20 (**d**) biologically independent samples. Asterisks or ND indicates significant or no significant difference in indicated comparisons, respectively.

mechanisms. This result suggests a mechanistic insight—revealing that oncogenic RNF43 induces its own expression by a positive feedback circuit of Wnt signalling, which we have reported previously[18], and then further suppresses the p53 pathway to facilitate tumorigenesis (Fig. 5d).

Our data suggest that *RNF43* mutations combined with activating mutations in Ras have the potential to fulfil not only two but all three steps of the traditional multi-step model of colon carcinogenesis (concurrent Wnt activation and p53 inactivation by RNF43 mutation and Ras activation by *KRAS* mutation[44–46]). To investigate the relevance of our findings to human cancer, we performed a comprehensive analysis of The Cancer Genome Atlas (TCGA) database to determine whether mutations in these genes affect patient outcome. This analysis confirmed that

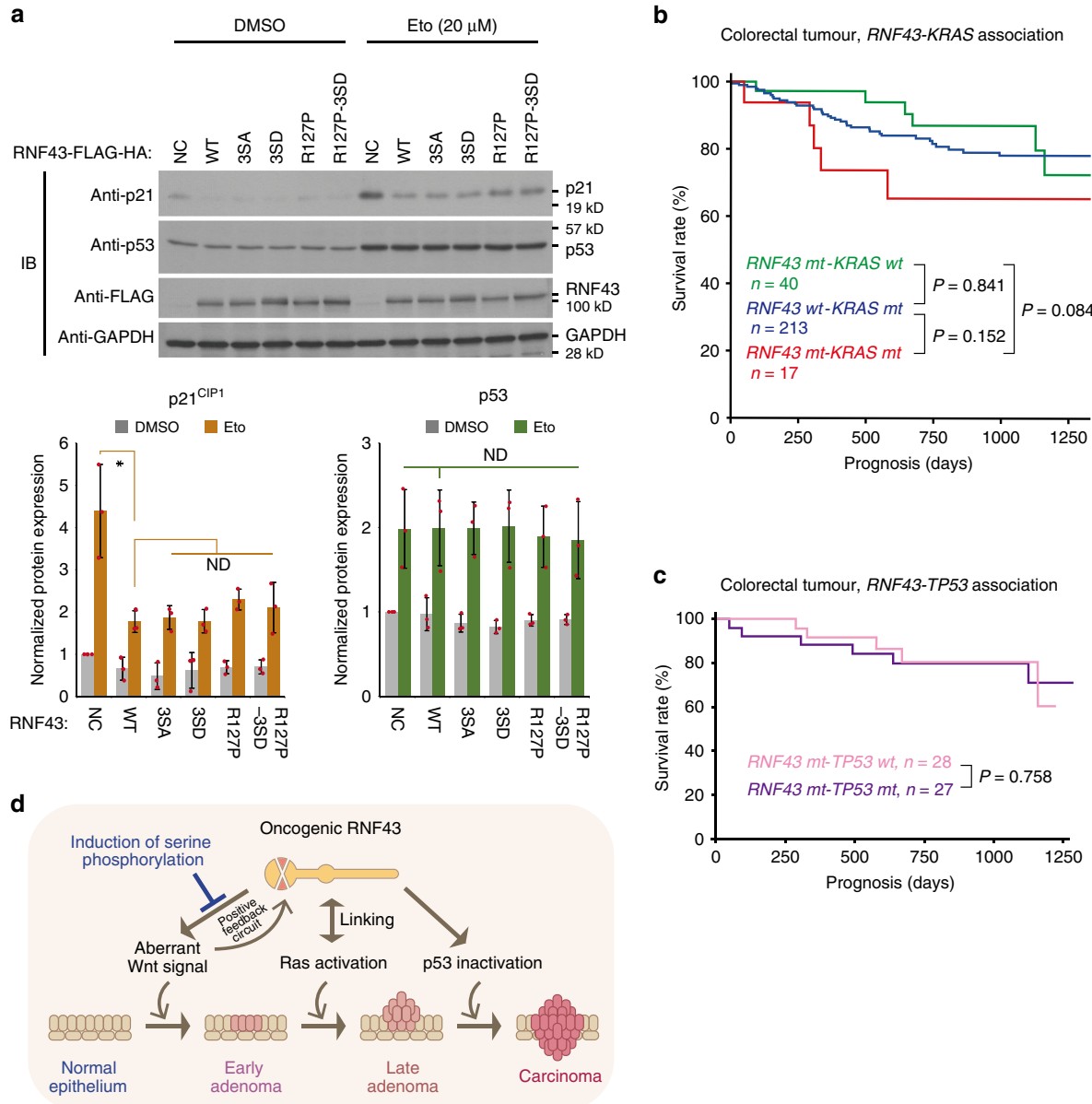

**Fig. 5 Cooperation of mutant RNF43 with active Ras establishes Wnt-Ras-p53 axis. a** Induction of p53 and p21 protein was examined by immunoblotting (IB) with Eto treatment. Expression of p53 and p21 in empty vector-transfected NC cells with DMSO treatment was set to 1. Bar graphs and error bars represent mean ± sd. Red circles indicate individual values of each sample. Asterisks indicate significant differences from NC cells stimulated with Eto ($P <$ 0.05, one-way ANOVA, $n = 3$ biological replicates). ND indicates no significant difference. **b, c** Prognosis of patients with colorectal tumour that carry genetic mutations in *RNF43* with or without *KRAS* (**b**) or in *RNF43* with or without *TP53* (**c**) is shown. Sample number and *P* value determined by log-rank test (**b, c**) with Holm adjustment for multiple comparisons (**b**) are indicated in each graph. **d** Schematic of biological role of RNF43 oncogenic mutations in multi-step tumorigenesis. Oncogenic RNF43 mutants that promote Wnt signalling and inhibit the p53 pathway cooperate with activating Ras mutations to complete all the steps of multi-step colorectal tumorigenesis.

mutations in *RNF43* or *ZNRF3* occur independently, supporting our theory that oncogenic *RNF43* and *ZNRF3* mutations act as dominant negatives[18] (Supplementary Fig. 9a). Furthermore, we found that co-occurrence of *KRAS* and *TP53* mutations is significantly associated with poor outcome (Supplementary Fig. 9b), as reported recently[47].

*RNF43* mutations were generally associated with poor outcome in patients with colorectal cancer, regardless of the microsatellite instability phenotype (MSI) status (Supplementary Fig. 9c). Importantly, Kaplan–Meier analysis revealed that co-occurrence of *RNF43* and *KRAS* mutations was associated with a poorer outcome for colorectal cancer, compared to single mutations of

each gene (Fig. 5b), although the samples size was not enough to obtain significant differences. In contrast, *TP53* mutations did not alter the outcome of colon cancer patients with *RNF43* mutations (Fig. 5c), supporting that RNF43 functions in the p53 pathway as shown in Fig. 5a and Supplementary Fig. 8. Moreover, colorectal tumours with mutations in *RNF43* lacked mutations in *APC*[48] and *TP53* (Supplementary Fig. 9d). These mutually exclusive data suggest that mutations in *RNF43* represent an alternative mechanism of both activating Wnt and disabling p53 during tumorigenesis, instead of independent multi-step mutations in *APC* and *TP53*. Overall, our findings provide insight into why the co-occurrence of mutations in two genes, *RNF43* and *Ras*, greatly

accelerate tumorigenesis: *RNF43-KRAS* mutations cooperate to establish the Wnt-Ras-p53 axis, supporting the multi-step model of colorectal carcinogenesis (Fig. 5d).

**Phosphorylation converts onco-RNF43 to a tumour suppressor**. Our data suggest that the 3SA and 3SD mutations abrogate and facilitate, respectively, RNF43-mediated ubiquitination of Fzd and inhibition of Wnt/β-catenin signalling (Figs. 1c, d, 6d, Supplementary Fig. 4j). To examine whether the phosphorylation of RNF43 can modulate the activity of distant oncogenic mutations in the PA domain or in the RING-finger domain of RNF43, we introduced the 3SA or 3SD mutations into RNF43(I48T), RNF43(R127P) and RNF43(H292R) backgrounds. We previously showed that the RNF43(I48T) and RNF43(R127P) mutations affects the subcellular localisation of RNF43 and impairs the function of RNF43[18]. Interestingly, the introduction of the 3SD, but not 3SA, mutations into the oncogenic RNF43 (I48T) and RNF43(R127P) backgrounds could partially rescue RNF43-mediated inhibition of Wnt signalling, as assessed by the STF-Luc assay (Fig. 6a, Supplementary Fig. 10a). In contrast, 3SD phosphomimetic substitution did not recover the RING domain-dead RNF43(H292R) mutation, which further supports the presence of a phospho-switch (Supplementary Fig. 10a). In addition, the 3SD substitution almost completely abolished the oncogenic activity of the RNF43(R127P) both in vitro and in vivo (Fig. 6b, c, Supplementary Fig. 10b). However, the introduction of 3SD did not alter the levels of RNF43(R127P) protein (Supplementary Fig. 7a) or its aberrant ER localisation that we reported previously[18] (Supplementary Fig. 10c). Thus, phosphomimetic mutations within the conserved serine triplet can revert diverse Rspo-independent dominant-negative RNF43 mutants back to functional negative regulators of Wnt signalling, irrespective of whether the oncogenic mutation is extracellular (I48T, R127P; Fig. 6a, Supplementary Fig. 10a) or intracellular (S478A; Fig. 2a). Further, our results suggest that forced phosphorylation or substitution of the conserved serine triplet might represent a promising therapeutic approach (Fig. 5d).

Based on these data, and our discovery that phosphorylation of RNF43 affects ubiquitination of Fzd5 (Supplementary Fig. 4j), we hypothesised that the R127P oncogenic mutation reduces RNF43-mediated ubiquitination of Fzd5, and that the 3SD mutations restore ubiquitination activity to the R127P mutant. To test this hypothesis, we again treated cells co-expressing Fzd5 and RNF43 mutants with bafilomycin to inhibit lysosome-dependent degradation and monitored the levels of ubiquitinated Fzd5. As expected, we found that cells co-expressing RNF43(R127P) displayed less ubiquitinated Fzd5 than cells co-expressing wild-type RNF43 (Fig. 6d). Strikingly, the RNF43(R127P-3SD) showed restored ubiquitination of Fzd5 relative to RNF43(R127P) (Fig. 6d). Thus, mislocalisation of RNF43(R127P) reduces RNF43-mediated ubiquitination of Fzd5, and the 3SD phosphomimetic substitution can restore the ubiquitination and tumour suppressor activity to RNF43(R127P). These findings reveal phosphorylation of RNF43 as a potential therapeutic target to restore the inhibitory role of RNF43.

## Discussion

Our work shows that phosphorylation of three conserved serines regulates the function and oncogenic potential of RNF43 by influencing the ubiquitination and subsequent lysosomal degradation of Fzd. Phosphorylation of RNF43 is required to negatively regulate canonical and non-canonical Wnt signalling during embryonic development and in adult stem cells. Dysregulation of RNF43 phosphorylation leads to a breakdown in homoeostasis and an increase in oncogenic activity. Thus, both multi-step

phosphorylation and RSPO-Lgr4/5/6 regulate RNF43 to control the surface level of Fzd and Wnt signalling activity (Fig. 6e). Furthermore, phosphorylation status seems to be the most critical regulator of RNF43, as it acts downstream of Rspo/Lgr5. Therefore, the phosphatase and the upstream signal that regulates RNF43 phospho-status should be identified to fully understand the mechanism of Wnt signalling regulation.

Importantly, combining phosphomimetic substitution with distant oncogenic RNF43 mutations (i.e. I48T and R127P) restores RNF43-mediated tumour suppression. The R127P mutation leads to mislocalisation of RNF43, which preclude phosphorylation at the conserved serine triplet. The 3SD mutation restored RNF43(R127P)-mediated ubiquitination of Fzd, apparently without restoring the localisation of this mutant. In addition, RNF43(R127P) exhibited a lower level of phosphorylation relative to RNF43(WT), suggesting that the function of RNF43 is not regulated directly by localisation, but by localisation-dependent phosphorylation. Our data thus indicate that RNF43 is a therapeutic target for patients harbouring oncogenic mutations outside the serine triplet and RING-finger domain. Suppressing RNF43 mutation-dependent tumorigenesis using Wnt inhibitors such as porcupine inhibitors (e.g., IWP-2) may be effective, because oncogenic RNF43 facilitates signalling activity in the presence of Wnt, as shown in our reporter assays, and we have already reported that tumorigenic hyperplasia of intestinal organs in the absence of both RNF43 and ZNRF3 is inhibited by IWP-2 treatment[38]. However, such a general inhibition of Wnt production affects a broad array of cell types and organs that maintain homoeostasis under the control of Wnt, regardless of RNF43 expression, and so may have serious side effects. Additionally, inhibiting the Rspo-Lgr module may not be effective for anti-cancer therapy, as we have previously found that oncogenic RNF43 does not require Rspo for the acceleration of Wnt signalling[18]. In contrast, the recovery of RNF43 activity by targeting serine phosphorylation could be a potential approach for tumour suppression with milder side effects, since it is predicted to only affect RNF43 mutant cells.

Recently, another research group observed the apparent hyperproliferation of the gastric mucosa in RING-dead RNF43 mutant mice, although these mice developed healthy intestines and did not exhibit neoplastic expansion of the ISC region, as reported previously in RNF43/ZNRF3 DcKO mice[11,30]. These results may indicate the importance of another function of RNF43, namely suppression of the p53 pathway. We demonstrate in Figs. 5a and 6d and Supplementary Fig. 8 that this suppression does not require the RING-finger domain-dependent ubiquitinating activity of RNF43, which is essential for the degradation of Fzd. RNF43 mutants that lack ubiquitination activity because of extracellular mutation (R127P) or a broken phospho-switch (3SA) retain the ability to suppress p53, whereas a lack of RNF43 expression causes a loss of suppression of both the Wnt and p53 pathways. Therefore, the ability of ubiquitination-dead mutants to suppress the p53 pathway may depend on the binding of RNF43 to p53 but not on phosphorylation, similar to the suppression of Wnt signalling by the nuclear RNF43-Tcf4 interaction[30]. In Supplementary Fig. 4i, we indeed demonstrate suppression of Wnt signalling via nuclear RNF43-Tcf4 binding, but at a marginal level relative to the mechanism associated with Fzd degradation at the surface. We previously reported that *RNF43* is a direct target of Wnt signalling; namely, a feedback loop is established (Fig. 5d), and mutations can further induce Fzd accumulation and p53 inhibition[18]. In this study, we demonstrated that a *RNF43* mutation can cooperate with *KRAS* to induce Wnt-RAS-p53 axis activity and thus drive tumorigenesis.

Furthermore, we demonstrated that mutations in the phospho-regulated serines that affect Wnt signalling did not alter RNF43-

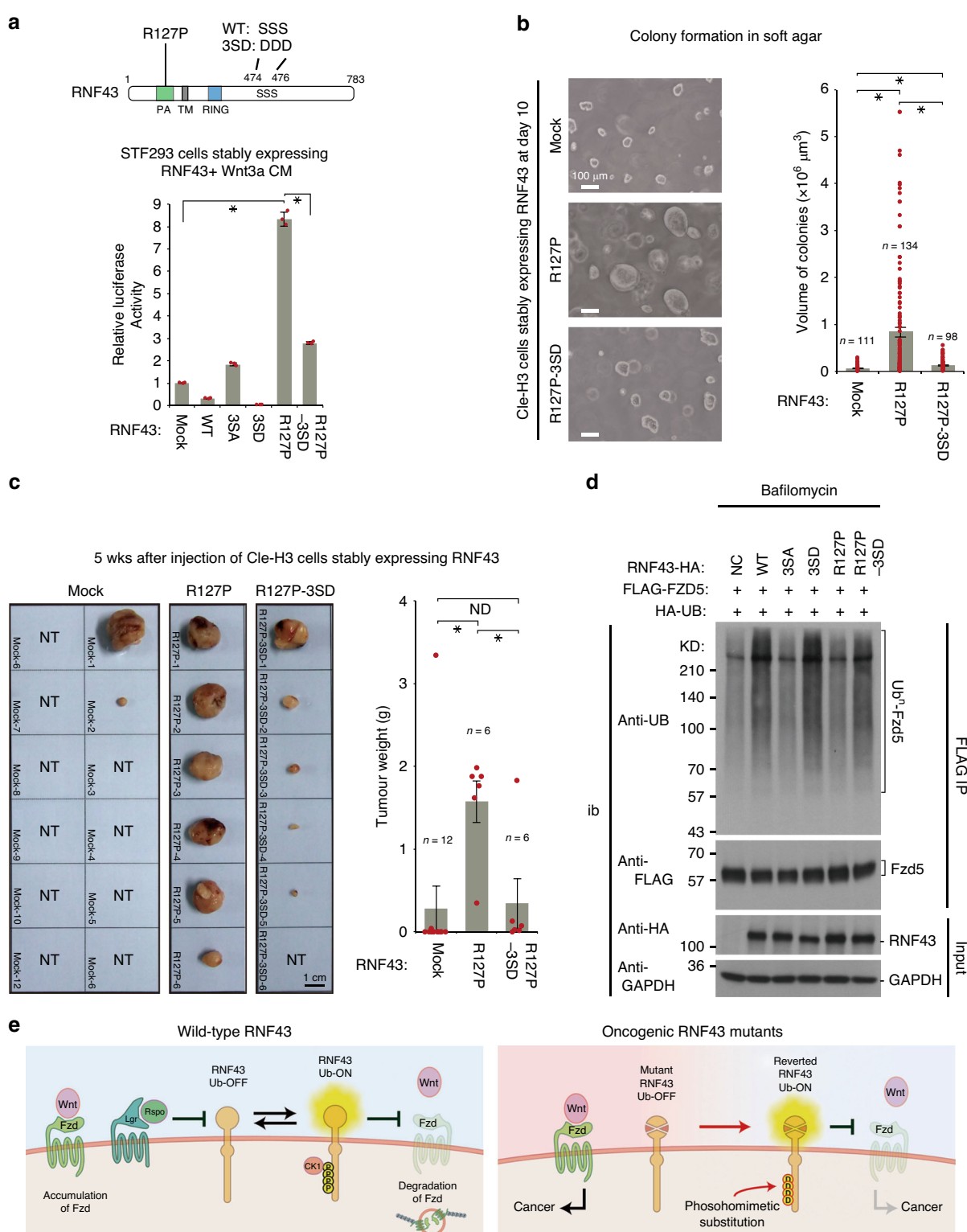

mediated p53 inhibition. Unfortunately, the number of patient tumour samples containing specific RNF43 missense mutations that reliably induce excessive Wnt signalling and/or maintain p53 inactivation is currently insufficient for database analysis (Supplementary Fig. 6a, b)[18]. Thus, we could not complete the analyses for the prognosis of colorectal tumour patients to clarify the roles of RNF43 phospho-regulation in tumorigenesis due to an insufficient numbers of samples.

The current databases do not classify correctly or in detail the type of mutations within a gene. It would be necessary to classify the type of RNF43 mutations as complete or partial deletions, or missense mutations, and to link them to functional changes, in order to fully understand the molecular and cellular roles for these mutations in tumorigenesis. This is especially so because RNF43 suppresses the p53 and Wnt signalling pathways by different mechanisms. It was recently reported that an RNF43

**Fig. 6 Serine phosphorylation reverts oncogenic RNF43 to a tumour suppressor. a** The role of serine phosphorylation was examined using STF-Luc assay in an RNF43(R127P) mutant background. Luciferase activity in mock-transfected cells was set to 1 (mean ± sd). Schematic of RNF43 mutants used in Figs. 5, 6, Supplementary Fig. 7, 9 is shown. Independent values of each sample are shown as red circles. Asterisks indicate significant differences ($P < 0.05$, one-way ANOVA, $n = 3$ biologically independent samples) from RNF43(R127P) cells. **b** Colony-forming activity was evaluated following expression of RNF43 phospho-mutant forms in Cle-H3 cells via soft agar assay and volume of colonies was estimated. Scale bars, 100 μm. Asterisks indicate significant differences from RNF43(R127P) tumour. **c** Tumour growth was examined in nude mice with Cle-H3 cells following the expression of RNF43 phospho-mutant forms at 5 wks after Cle-H3 injection and tumour weight was measured. Images for all of the tumours are shown. Scale bar, 1 cm. NT indicates no tumour observed. Bar graphs and error bars in (**b**, **c**) represents mean ± sem of biologically independent samples. Red circles indicate individual values of each sample. The $P$ values for the indicated comparisons were determined by one-way ANOVA ($P < 0.05$). $n = 98–134$ (**b**), $n = 6–12$ (**c**) biologically independent samples. Asterisks or ND indicates significant or no significant difference in indicated comparisons, respectively. **d** Ubiquitination of Fzd5 by RNF43 phospho-mutants was examined with bafilomycin A₁ by immunoprecipitation (IP)-IB experiments. **e** Schematic of molecular mechanism and biological role of RNF43 phosphorylation in Wnt signalling and multi-step tumorigenesis. Wild-type RNF43 is activated by serine phosphorylation. Oncogenic RNF43 with R127P extracellular mutation is reverted to a functional tumour suppressor via the introduction of phosphomimetic mutation.

(G659fs) mutation is frequently associated with colorectal tumours having a MSI phenotype[48,49] and a better prognosis. However, we do not have evidence suggesting that this fs mutant protein would not be expressed by nonsense-mediated mRNA decay (NMD) or that it would function normally as reported recently[50,51]. Indeed, not all deletion or missense mutations exhibiting a high-score (frequency of the mutation or impact expected on the protein structure) in database analyses converted RNF43 to a dominant-negative-Wnt activating form, as shown in our previous[18] and current study (Supplementary Fig. 6a), and MSI did not markedly affect the prognosis of patients harbouring the RNF43 mutation in our analysis (Supplementary Fig. 9c). Our data indicate that many cytoplasmic mutations are passengers, except those in the RING domain and the phospho-switch, whereas mutations in the PA domain confer an oncogenic effect.

Our careful analysis of missense mutations in RNF43 has provided significant insights into the operation of this tumour suppressor with three functional domains: the extracellular PA domain for interaction with Fzd or Rspo, RING domain for ubiquitination and a phospho-switch for functional control. We also identified a critical role of the phospho-switch by demonstrating its epistatic control of mutations in the PA domain. Our findings suggest that therapeutic phosphomimetics could revert an oncogenic RNF43 mutant to a functional tumour suppressor.

## Methods
**Accession numbers of RNF43 and ZNRF3.** Human RNF43, NP_001292473. Human ZNRF3, NP_001193927. Mouse Dvl2, NP_031914. Mouse Fzd5, NP_001036124. Mouse RNF43, NP_766036, Naked-mole rat RNF43, XP_021104324. Bengalese finch RNF43, XP_021396062. Three-toed box turtle RNF43, XP_026514312. Tropical clawed frog RNF43, XP_002935238. Zebrafish RNF43, XP_021332049.

**Plasmids.** The mammalian expression vectors pcDNA3-hRNF43(WT, Δ366-441, Δ366-478, I48T, R127P)-HA, pCS2 + -FLAG-mFzd5 and pCS2-myc-mDvl2 were described in our previous reports[18]. A series of deletion mutants, pcDNA3-hRNF43(ΔSRR1, ΔSRR2, ΔSRR1/2, ΔSRR2-1, ΔSRR2-2, ΔDvl-(N, C), ΔSSS, S476del)-HA, and missense mutants, pcDNA3-hRNF43 and pcDNA4/TO-hRNF43 (S474A, S474P, S475A, S476A, S474–476A (3SA), S474–476D (3SD), S474–476E (3SE), S474–476T (3ST), S478A (SA), S478D (SD), S478E (SE), S478P (SP), S474–476A:S478D (3SA-SD), S474–476D:S478A (3SD-SA), H292R, S539A, E318D, Q344H, Y357C, R389H, L418M, G447E, V479L, T483M, S532F, S532D, R519Q, E541K, H549N, R554G,P569H, P587S, R600S, S607L, D628G, E662K, W726L, I48T:3SA, I48T:3SD, R127P:3SD and R127P:3SA-HA), were generated using polymerase chain reaction (PCR). PCR products amplified using KOD FX Neo polymerase (KFX-201, Toyobo) together with pcDNA3-hRNF43-HA or pcDNA4TO-hRNF43-x2FLAG-HA templates and the relevant primers (all primers used in this study to generate expression vectors for RNF43 derivatives are described in Supplementary Methods) were then self-ligated using a ligation kit (6022, DNA Ligation Kit Ver.2.1, Takara). All the expression plasmids generated were sequenced to confirm identity. Wild-type and missense mutants of hRNF43 (3SA, 3SD, 3SE, S478A, R127P and R127-3SD) bearing HA or x2 FLAG-HA tags were inserted into pMX-puro vectors for the derivation of stable cell lines expressing RNF43 constructs.

**Cell culture, transfection and reagents.** Cell lines into which the SuperTopFlash Wnt reporter had been introduced (HEK293, termed STF293 following introduction of the reporter), HEK293, HeLa, HCT116, Platinum-E (Plat-E), Platinum-A (Plat-A), MB352, Rspo1/HEK293 and Cle-H3 were grown in DMEM (D5796, Sigma) supplemented with 10% foetal bovine serum (10270, Gibco). NIH3T3 cells were grown in DMEM supplemented with 10% bovine serum (16170, Gibco). STF293 cells were described in our previous report[18,52]. Cle-H3 cells were provided from RIKEN BRC. Wnt3a/L cells were the kind gift of S. Takada (NIBB, Japan) and were grown in DMEM/F-12 HAM (D8062, Sigma) supplemented with 10% foetal bovine serum. Control and Wnt3a-conditioned media (Wnt3a CM) were obtained from 24 h culture of these cells. Plasmids were transfected using FuGENE HD Transfection Reagent (E231A, Promega) according to the manufacturer's protocol. For retrovirus-mediated gene transduction, NIH3T3, Cle-H3, HCT116 or STF293 cells were infected with retroviruses produced in Plat-E or Plat-A packing cells[53]. These cells were then cultured in the presence of 5 μg/ml puromycin (P8833, Sigma) for 1 week. Stable expression of RNF43 constructs was confirmed by immunoblot analysis. CHIR-99021 (3 μM; SML-1046, Sigma) and IC261 (2 μM; ab145189, Abcam) were used to inhibit the kinases GSK-3β and CK1 respectively. The iCRT3 (50 μM; SML-0211, Sigma) was used to inhibit the binding between β-catenin and Tcf/Lef. Etoposide (20 μM, VP-16, E1383, Sigma) was used for 12 h to induce p53-dependent p21 expression.

**Luciferase assays.** STF293 cells stably expressing HA-tagged RNF43 WT or mutant forms or STF293 control cells were seeded into 24-well plates ($5 × 10^4$ cells) and transfected with RNF43 expression plasmids using FuGENE HD Transfection Reagent. Wnt3a CM (1/4 of total volume) was added to the culture medium 24 h after transfection (STF293 cells) or after seeding cells (RNF43-expressing stable cell lines) before cells were cultured for an additional 24 h. Cells were harvested and lysed in 100 μl of cell culture lysis reagent. The luciferase activity was measured with the Luciferase Assay System (E1501, Promega) using 10 μl of lysate and 50 μl of luciferase assay substrate. Luminescence was quantified with a luminometer (GLOMAX 20/20 LUMINOMATER, Promega). The relative level of luciferase activity in empty vector- or RNF43(WT)-transfected cells, or in mock-transfected cells that had undergone Wnt3a stimulation was set to 1. All experiments were repeated independently three times or more.

**Immunoprecipitation and immunoblotting.** Cells expressing RNF43 constructs were lysed with IP lysis buffer containing 50 mM Tris-HCl (pH 7.6), 150 mM NaCl, 0.7% Triton X-100, 0.4 mM Na3VO4, 0.4 mM EDTA, 10 mM NaF and 10 mM sodium pyrophosphate and cOmplete Mini EDTA-free (19541400, Roche). Lysates were incubated on ice for 20 min and then centrifuged at 16,000g for 20 min at 4 °C. After determination of protein concentration via the Bradford assay (500-0006, Protein Assay, Bio-Rad), cell lysates (10 μg/lane) were subjected to SDS-PAGE on 8–10% acrylamide gels in the presence or absence of 50 μM Phos-tag Acrylamide (AAL-107, FujiFilm-Wako) and separated proteins were transferred to an Immobilon-P membrane (IPVH00010, Millipore). The membranes were probed with antibodies against HA (HA.11-16B12, MMS-101R, Covance) at 1:5000 dilution, Myc (9E10, PRB-150P, Covance) at 1:5000 dilution, FLAG (F3165, M2 and F4042, M5, Sigma) at 1:5000 dilution, active β-catenin (4270 and 8814, D13A1, Cell signaling Technology) at 1:1000 dilution (both), p53 (sc-126, DO-1, Santa Cruz Biotechnology) at 1:1000 dilution, p21 (sc-6246, F-5, Santa Cruz Biotechnology and 64016, 2947, Cell signaling Technology) at 1:250 and 1:1000 (both) dilution, Bax (5023, D2E11 and 14796, D3R2M, Cell signaling Technology) at 1:250 and 1:1000 dilution, c-myc (sc-764, N-262, Santa Cruz Biotechnology) at 1:400 dilution, ubiquitin (sc-8017, P4D1, Santa Cruz Biotechnology) at 1:1000 dilution, HH2B (sc-10808, FL-126, SantaCruz Biotechnology) at 1:500 dilution, IRE1α (3294, Cell

signaling Technology) at 1:1000 dilution and GAPDH (016-25523, 5A12, FujiFilm-Wako) at 1:10000 dilution in TBST buffer containing 20 mM Tris-HCl pH8.0, 150 mM NaCl and 0.05% Tween 20. Immune complexes were detected either with HRP-conjugated secondary antibodies against mouse IgG (W4021, Promega), Mouse TrueBlot Ultra (18-8817-33, Rockland), rabbit IgG (W4018, Promega) or with Rabbit TrueBlot Ultra (18-8816-31, Rockland) and either Pierce Western Blotting Substrate (NCI3106, Thermo Fisher Scientific) or Immobilon Western Chemiluminescent HRP Substrate (WBKLS0050, Millipore). For co-immunoprecipitation experiments, cell lysates were incubated with the indicated antibodies for 8–12 h followed by incubation with either protein A-Sepharose beads (GE Healthcare Bioscience) for 1 h, or with anti-FLAG-M2 affinity gel (A2220, Sigma) or anti-HA agarose beads (A2095, Sigma) for 8–12 h. Beads were washed 5 times with IP lysis buffer and the retained proteins were eluted by incubation with 200 µg/ml 3xFLAG peptide for 20 min at room temperature (F4799, Sigma) or by boiling with SDS sample buffer for immunoblot analysis. Protein levels were quantified with densitometry using ImageJ software (NIH) and normalised to an internal loading control (GAPDH).

**2D-PAGE**. Cells expressing RNF43 constructs were lysed with 2D lysis buffer containing 60 mM Tris-HCl (pH8.8), 5 M urea, 1 M Thiourea, 1% CHAPS, 1% Triton-X100, 10 mg/ml DTT, 0.4 mM Na₃VO₄, 0.4 mM EDTA, 10 mM NaF and 10 mM sodium pyrophosphate and cOmplete Mini EDTA-free (19541400, Roche) for 20 min on ice. Supernatant was collected following centrifugation at 15,000 $g$ for 20 min at 4 °C and incubated with iodoacetoamide (final concentration, 100 mM) for 10 min at room temperature. Separation of proteins by isoelectric focusing (IEF) electrophoresis in the first dimension was performed with AgarGEL (pH 10-3 or 8-5, A-M310 or A-M58, Atto) and DiscRun-R (300 V, 3.5 h) (WSE-1500, Atto) according to the manufacturer's protocol. The pH-separated proteins were then separated sequentially depending on their molecular size in the second dimension using e-PAGEL (7.5%) (E-D7.5 L, Atto). Separated proteins were subsequently subjected either to autoradiography or to immunoblot analysis using antibodies against HA.

**Phosphate depletion culture and ³²Pᵢ metabolic labelling**. Depletion of phosphate and ³²Pᵢ metabolic labelling of RNF43 were performed as described below. Cells expressing exogenous RNF43 or carrying HA-tagged endogenous RNF43 were incubated under normal conditions for 42–48 h, washed twice before culture with sodium phosphate-free DMEM (11971, Gibco) containing 10% diarised FCS (04-311-113, Biological Industries) and then cultured for 2 h to deplete phosphate from cellular proteins[54]. These cells were either directly subjected to 2D-PAGE and immunoblot experiments or underwent metabolic labelling of RNF43. For metabolic labelling, phosphate-depleted cells were subsequently cultured in phosphate-depleted culture media containing 1 mCi/ml of [³²P] orthophosphoric acid (³²Pᵢ) (NEX053S, Parkin Elmer) for 4 h in order to label cellular proteins. After extensive washing, labelled cells were lysed with IP lysis buffer and underwent immuno-precipitation as detailed above using anti-HA agarose beads or anti-FLAG affinity gel (A2095 or A2220, Sigma). Immunoprecipitates were eluted with SDS sample buffer or 2D lysis buffer for SDS- or 2D-PAGE, respectively, and subjected to autoradiography and quantitative analysis with a BAS-4000 image analyser and Multi Gauge software Ver. 3.0 (Fuji Film).

**In vitro kinase assays**. Cells expressing RNF43 constructs were lysed with IP lysis buffer containing 50 mM Tris-HCl (pH 7.6), 150 mM NaCl, 0.7% Triton X-100, aprotinin (10 µg/ml), leupeptin (10 µg/ml), 10 mM iodoacetamide and 1 mM PMSF but without the addition of phosphatase inhibitor and then underwent immunoprecipitation as detailed above using anti-HA agarose beads. Immuno-precipitants were washed 6 times for 20 min per wash with high-salt wash buffer containing 50 mM Tris-HCl (pH 7.6), 500 mM NaCl, 1.0% Triton X-100, aprotinin (10 µg/ml), leupeptin (10 µg/ml), 10 mM iodoacetamide and 1 mM PMSF to eliminate RNF43 interacting proteins.

Immunoprecipitants were washed twice with kinase reaction buffer containing 50 mM Tris-HCl (pH 7.5), 10 mM MgCl₂, 0.1 mM EDTA, 2 mM DTT, 0.01% Brij 35 and then incubated in kinase reaction buffer supplemented with 100 IU CK1 or CK2 (P6030 or P6010, NEB), 10 µCi [γ-³²P] ATP (NEG502A, Parkin Elmer) and 10 nM ATP for 30 min at 30 ˚C. Phosphorylated immunoprecipitants were washed 4 times with IP lysis buffer and RNF43 proteins were eluted using SDS sample buffer before being subjected to SDS-PAGE and autoradiography. Quantitative analysis was performed with a BAS-4000 image analyser and Multi Gauge software Ver. 3.0 (Fuji Film).

**Identification of phospho-proteins by LC–MS/MS analysis**. STF293 cells stably expressing RNF43 were lysed with IP lysis buffer 48 h after seeding. Lysates underwent immunoprecipitation as detailed above using anti-HA agarose beads before being washed 7 times and RNF43 eluted by incubation using 100 µg/ml HA peptide (I2149, SIGMA) for 20 min at room temperature. Eluted proteins were separated by SDS-PAGE on 8% acrylamide gels before undergoing silver staining. The band corresponding to RNF43 (~100 kDa) was removed from the stained gel and the protein therein was subjected to in-gel digestion with trypsin or chymo-trypsin. Identification of proteins was performed using our standard protocol[55] as

described below. Resulting peptides were dissolved in a solution containing 0.1% trifluoroacetic acid and 2% acetonitrile and analysed by an LTQ Orbitrap Velos Pro mass spectrometer (Thermo Fisher Scientific, Waltham, MA) coupled with a nanoLC instrument (Advance, Michrom BioResources, Auburn, CA) and HTC-PAL autosampler (CTC Analytics, Zwingen, Switzerland). Peptide separation was performed with an in-house pulled fused silica capillary (internal diameter, 0.1 mm; length, 10 cm; tip internal diameter, 0.05 mm) packed with 3-µm C18 L-column (Chemicals Evaluation and Research Institute, Japan). The mobile phases consisted of 0.1% formic acid (A) and 100% acetonitrile (B). Peptides were eluted with a gradient of 5_35% B for 40 min at a flow rate of 300 nL/min. Collision-induced dissociation (CID) spectra were acquired automatically in the data-dependent scan mode with the dynamic exclusion option. Full MS spectra were obtained with Orbitrap in the mass/charge (m/z) range of 300-2000 with a reso-lution of 60,000 at m/z 400. The 12 most intense precursor ions for in the full MS spectra were selected for subsequent ion-trap MS/MS analysis with the automated gain control (AGC) mode. The AGC were set to $1.00 \times 10^6$ for full MS, $1.00 \times 10^4$ for CID MS/MS. The normalised collision energy values were set to 35%. Lock mass function was activated to minimise mass error during analysis. The peak lists were generated by MSn.exe (Thermo Fisher Scientific) with a minimum scan/group value of 1 and were compared with IPI_Human database using the Mascot for determining serine, threonine and tyrosine phosphorylation on RNF43.

Original dataset of the analyses has been deposited in the ProteomeXchange Consortium (http://proteomecentral.proteomexchange.org/) via the via the jPOST partner repository under dataset identifiers PXD020598 and PXD02059.

**Fzd-ubiquitination analysis**. Expression vectors of HA-tagged Ub, Fzd5 and RNF43 mutants were transfected into STF293 cells with FuGENE HD. Cells were cultured for 36 h after transfection and then treated with bafilomycin A1 (15 nM, 12 h, BVT-0252, Adipogen). Cellular Fzd5 proteins were immunoprecipitated in RIPA buffer containing aprotinin (10 µg/ml), leupeptin (10 µg/ml), 10 mM iodoacetamide, 1 mM PMSF, 0.4 mM Na3VO4, 0.4 mM EDTA, 10 mM NaF and 10 mM sodium pyrophosphate with anti-FLAG(M2) beads (A2220, Sigma). The ubiquitination status of Fzd5 protein was examined by immunoblot analysis with the indicated antibodies.

**Flow cytometry**. Cells expressing RNF43 constructs were cultured in the presence or absence of small molecule kinase inhibitors (GSK-3βi, 3 µM CHIR-99021 for 24 h or CK1i, 2 µM IC261 for 4 h) or Rspo (10 ng/ml for 3 h) and then harvested using PBS containing 1 mM EDTA and resuspended in FACS staining buffer (PBS containing 0.1% BSA and 0.02% sodium azide). Single-cell suspensions of STF293 cells ($1 \times 10^5$ cells) were stained for 45 min on ice with a combination of anti-panFzd (OMP-18R5, kind gift of A. Gurney, Oncomed)[56] at 1:100 dilution and anti-human IgG-FITC (109-095-098, Jackson Immunoresearch) antibodies at 1:250 dilution. All analyses were performed using a FACSCalibur flow cytometer and CellQuest Ver. 3.3 (Becton Dickinson, BD) software. All graphs are presented with normalised scales for every histogram. All FACS data in this study were acquired and displayed with the same strategy shown in Supplementary Fig. 1g.

**Generation of RNF43-knockin and -knockout cells by CRISPR**. Genomic editing of the RNF43 locus was performed using the CRISPR/Cas9 technique and the GeneArt CRISPR Nuclease Vector Kit (A21174, Life Technologies). The nucleotide sequence encoding the full-length HA amino acid tag (YPYDVPDYASLGGP) was inserted in-frame at the C-terminal end of the RNF43 genomic locus before the stop codon in STF293 cells. Moreover, STF293 cells were subjected to either deletion of the conserved serines in the phospho-switch (SSS) or the introduction of an established tumorigenic R127P mutation. Template ssDNAs and plasmids used to express guide RNAs were transfected into the cells with Lipofectamine 3000 reagent. OFP-expressing, genome-edited cells were isolated at 24 h post-transfection via cell sorting on a FACSAria II flow cytometer with FACSDiva software Ver. 8.0 (BD), cloned by limiting dilution and screened by PCR. All genome-edited cells were sequenced to confirm the introduction of mutations. The oligonucleotide sequences for the guide RNAs, template ssDNAs and primers used in PCR screening and sequencing are detailed in Supplementary Fig. 3 and described in the Supplementary Methods.

**Cellular localisation analysis**. Localisation of RNF43 mutants was examined with the expression vectors for RNF43(WT, 3SA and 3SD)-EGFP or RNF43(WT, 3SA, 3SD, R127P and R127P-3SD)-HA with anti-HA-Alexa488 antibodies (A488-101L, Covance) in HeLa cells or with Anti-HA antibodies and Alexa488 Tyramide SuperBoost kit (B40912, Invitrogen) in HA-KI STF293 cells. Cells were fixed with 2% formalin, stained according to manufacturer's standard protocol, then images of cells were taken with BX51 fluorescent microscope, DP71 camera, DP Controller Ver. 3.1.1.267 and DP Manager software Ver. 3.1.1.208 (Olympus).

**Fractionation of cellular proteins**. Cellular fractions of RNF43-KI STF293 cells were suspended in a separation buffer (10 mM Hepes (pH 7.9), 10 mM KCl, 1.5 mM MgCl₂, 0.5 mM DTT, 0.25 M sucrose, phosphatase inhibitors and protease inhibitors) and lysed with a Dounce homogeniser by 40 strokes. The cell lysates were separated into nuclear (500$g$, 5 min), heavy microsome (HM) membrane

(12,000$g$, 20 min) and cytosolic fractions (supernatant). ER membrane was collected by IP with Anti-IRE1α antibodies (#3294, 14C10, Cell Signaling Technology) at 1:400 dilution. Endogenous RNF43 protein was concentrated from each cellular fraction via IP with anti-HA antibodies in a cell lysis buffer. Subcellular localisations of RNF43 were confirmed by immunoblotting using the antibodies indicated.

**Zebrafish maintenance and injection into zebrafish eggs**. Wnt reporter zebrafish (AB, OTM:d2EGFP-transgenic fish) were raised and maintained under standard conditions[57]. Experimental zebrafish care was performed in accordance with institutional (Gunma and Osaka University) and national guidelines and regulations. For all injections, 50–150 pg mRNA encoding human RNF43 was injected into zebrafish eggs at the one-cell stage.

**Quantitative PCR (qPCR)**. Total RNA was isolated from HCT116, STF293 cells or staged OTM:d2EGFP-transgenic zebrafish embryos homogenised in Trizol reagent (15596018, Invitrogen). 1 μg RNA was used for cDNA synthesis using the ReverTra Ace qPCR RT Master Mix with gDNA Remover (FSQ-301, Toyobo). Quantitative real-time PCR (qPCR) for *d2EGFP*, *tbx6*, *axin2*, *nkd1*, *znrf3* and *β-actin* or *hCDKN1A*, *hBAX*, *hZNRF3* and *hGAPDH* (primers detailed in Supplementary Method) was performed using standard PCR conditions on a StepOnePlus Real-Time PCR System (Applied Biosystems) or Mx3000P Real-Time QPCR System (Agilent) with Power SYBR Green PCR Master Mix (4367659, Applied Biosystems) or TUNDERBIRD SYBR qPCR Mix (QPS-101, Toyobo). Expression levels were normalised to *β-actin* expression. All experiments were performed in triplicate. The sequence of all primers used for qPCR in this study are detailed in Supplementary Methods.

**Whole mount in situ hybridisation**. Digoxigenin-labelled RNA antisense probes for in situ hybridisation were generated via in vitro transcription using plasmids containing full-length cDNAs for *myod1*[58], *ntla*[59] and *dlx3b*[60] according to the manufacturer's protocol (DIG RNA labelling kit; 11175025910, Roche Life Science). Whole mount in situ hybridisation was performed according to a standard protocol. Zebrafish embryos were fixed in 4% paraformaldehyde in PBS, antisense probes hybridised and stained with BM purple (11442074001, Roche Life Science).

**Organoid experiments**. Small intestine crypts were isolated from Vil-creERT2 mice and organoids were established using general procedure[61]. Mouse intestine washed with cold PBS, segmented, and removed villus by scraping. Then, crypts isolated using Gentle Cell Dissociation Reagent (07174, STEMCELL technologies). After counting isolated crypts, 100 crypts embedded in Matrigel (356255, Corning) with supplement of growth factors. Medium refreshed every other day. Organoids were maintained in organoid culture conditioned medium for retrovirus-mediated gene transduction. Retrovirus containing human RNF43, 3SD or 3SA constructs were produced in Plat-E cells[53] (RV-101, Cell Biolabs). Retrovirus containing media was concentrated using Retrovirus Concentrator (631456, Takara) and introduced to organoid cultures in the standard condition[61]. Organoid cultured in the presence of Wnt3a and Nicotinamide 3 days before infection. On the infection day, organoids fragmented by mechanical dissociation followed by chemical dissociation using TripLE (12605-010, Invitrogen) at 37 °C. Fragmented organoid combined with retroviral solution in the presence of polybrene (H9268, Sigma) and Y-27632 (Y0503, SIGMA) for spinoculation at 32 °C, 600$g$, for 1 h. After spinoculation, plate incubated additional 6 hrs at 37 °C. Infected organoid fragments collected and embedded in Matrigel with culture medium containing Y-27632 (Y0503, Sigma). Infected organoids were selected from 3 days post-infection for 1 week with 2.5 μg/ml puromycin in order to remove uninfected organoids.

Expression of RNF43 proteins was induced by treating transduced organoids with 1 μM 4-hydroxytamoxifen (4-OHT, T5648, Sigma) for 6 h at 37 °C directly after passaging. Culture medium was then changed to ENR (EGF-Noggin-Rspo) medium containing either 1% or 10% Rspo as indicated (EGF, 50 ng/ml, PMG8043, ThermoFisher; Noggin, 100 ng/ml, 250-38, Peprotech; Rspo CM, produced by HEK293 cells). Organoids were imaged using an EVOS FL system (Life Technologies) and quantified on days 1, 2 and 3 following 4-OHT treatment in order to determine the ratio of dead:live organoids.

**Soft agar assay**. $1 \times 10^5$ NIH3T3 or Cle-H3 cells expressing RNF43 constructs were grown in DMEM containing 0.35% low melting temperature agarose (50101, Lonza) and either 10% CS or FCS, respectively. Cells were cultured in agar for 10 days in 6-well plates. Colonies formed under anchorage-independent conditions were imaged using an DP-12 system (Olympus) and quantified on day 6 or 10 of culture. The minor axis (S) and major axis (L) of each colony were measured at day 10 and colony volume was calculated as $V = (S^2 \times L)/2$.

**Sphere forming assay**. $5 \times 10^3$ Cle-H3 cells expressing RNF43 constructs were cultured on ultra-low attachment 24-well culture plates (3473, Corning) in DMEM supplemented with 10% FCS for 7 days. The cell spheroids formed under anchorage-independent conditions were imaged on day 7 of culture using a DP-12 system (Olympus) and spheroid growth was estimated via MTS assay (G3518, CellTiter 96 AQueous One Solution Cell Proliferation Assay, Promega) according to the manufacturer's protocol. Briefly, spheroids were incubated for 1.5 h with

detection reagent (100 μl reagent in 500 μl culture medium) before absorbance at 490 nm was measured using a spectrophotometer (SmartSpec 3000, Bio-Rad).

**In vivo tumorigenesis assays**. $1 \times 10^6$ (HeLa-, NIH3T3-RNF43) or $5 \times 10^5$ (Cle-H3-RNF43) cells were suspended in 50 μl of PBS and 50 μl of Matrigel (356230, growth factor reduced, Corning) before lateroabdominal injection of the cell suspension (100 μl/mouse) into nude mice (BALB/c$^{nu/nu}$). Injection of each set of cells expressing an RNF43 transgene into an individual mouse (right flank) was accompanied by injection of control cells (left flank). Tumour size was measured weekly and the tumour volume was calculated as $V = (S^2 \times L)/2$, as in soft agar assays. At the endpoint of the experiment, mice with tumours were sacrificed and the weight of the tumours was measured. All experimental animal care was performed in accordance with institutional and national guidelines and regulations. Experimental design using mice was approved by NATIONAL UNIVERSITY CORPORATION HOKKAIDO UNIVERSITY PROVISIONS ON ANIMAL EXPERIMENTS.

**Immunostaining of allografts**. Tumours arising from Cle-H3 cells with or without the expression of RNF43(3SA) mutant were fixed with 10% formalin and subsequently sectioned at 7 μm using a cryostat (HM550-VPD, Thermo Fisher Scientific). Sections were treated with 0.1% Triton-X in PBS for permeabilization and then with LAB solution (Polyscience, #24310) for antigen retrieval. Each sample was stained with anti-β-catenin mouse IgG1 (BD-TDL #199220) at 1:500 dilution, anti-Vimentin mouse IgM (Sigma, #V5255) at 1:2000 dilution and DAPI (1 μM) in combination with anti-mouse IgG1-Alexa555 (Invitrogen, #A-21127) at 1:1000 dilution and anti-mouse IgM-Alexa488 (Invitrogen, #A-21042) at 1:1000 dilution in 1.5% NGS and 0.1% BSA in TBST after blocking with 5% NGS in TBST. Tumour images were taken using a confocal laser scanning microscope (Carl Zeiss, Axio Imager Z1 & LSM700) equipped with a water-immersion ×40 objective lens (C-Apochromat 40 × /1.20 W Corr M27) and ZEN Black 2011 software (Zeiss). Z-stack images were processed and arranged using ImageJ software (1.52 v, NIH) and Photoshop CS5 (12.0 × 64, Adobe).

**TCGA database analysis**. All TCGA analysis was performed on The Cancer Genome Atlas website with datasets of patients with any tumour type (all tumour) or colorectal tumour containing mutations in RNF43, KRAS and/or TP53. All analyses were performed on the public website, The Cancer Genome Atlas (https://cancergenome.nih.gov) and GDC Data Portal (https://portal.gdc.cancer.gov).

*Statistics and reproducibility*. All $P$ values between samples in all experiments were determined via one-way analysis of variance (one-way ANOVA), or log-rank test on GDC Data Portal (https://portal.gdc.cancer.gov). Error bars represent standard deviation (sd) or standard error of the mean (sem) as indicated. All raw data and exact $P$ values in the analyses of this study are shown in Source Data file. The reproducibility of each experiments is shown as number of repeated/number of similar results. Supplementary Fig. 5a showed $n = 3/3$ reproducibility. Supplementary Figs. 2a–c, g, j, 3g, 4a–d,f–h, 7d, 10c showed $n = 2/2$ reproducibility. Experiments in Figs. 2f, 3a,d,e, 4j, 6d, 7a, 8a–c, 10b,c were preformed once. The result in Fig. 6d and Supplementary Fig. 4j directly supports each other. Result in all experiments that were not repeated was highly consistent in fact/theory with the results from other experiments in this or in our past study.

**Website**. All mutations in the human RNF43 gene were retrieved from the Catalogue of Somatic Mutations In Cancer database (http://cancer.sanger.ac.uk/cosmic/), The Cancer Genome Atlas (https://cancergenome.nih.gov) GDC Data Portal (https://portal.gdc.cancer.gov) and cBioPortal database (http://www.cbioportal.org).

**Reporting summary**. Further information on research design is available in the Nature Research Reporting Summary linked to this article.

## Data availability

The mass spectrometric datasets used in Supplementary Fig. 2h have been deposited in the ProteomeXchange Consortium (http://proteomecentral.proteomexchange.org/) via the via the jPOST partner repository under dataset identifiers PXD020598 and PXD02059. All full scan images of our blotting data used in this study are shown in Source Data file. Source data are provided with this paper.

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

## Acknowledgements

The authors thank M. Uchiumi, Y. Matsuzaki, M. Fujioka and S. Sakata for administrative and technical assistance. We also thank S. Takada, T. Oikawa, T. Kitamura and A. Gurney for providing materials, M. Oda for data analysis, Y. Yamaguchi and N. Kato for suggestions. This work was supported by KAKENHI to T.T. (25430102, 16K07105, 19K07633), Y.F. (16H06227), H.T. (15H04701, 17K19578), Y.O. (17H04016), T.I.

(16H05141, 26114006), K.N. (17H06301) and S.H. (15H04690, 18H02607) by the Ministry of Education, Culture, Sports, Science and Technology in Japan, JST CREST to M.M. (JPMJCR15G4), the Japan Foundation for Applied Enzymology and the Pancreas Research Foundation of Japan to T.T., and by the European Research Council (ERC (639050) and the Interpark Bio-Convergence Center Grant Program to B-K.K. This work was partly performed with the collaborative Research Project Program of the Medical Institute of Bioregulation, Kyushu University, with the joint research program of the Institute for Molecular, Cellular Regulation, Gunma University and with the Grant for Joint Research Project of the Research Institute for Microbial Diseases Osaka University.

## Author contributions

T.T. performed most of the biochemical, cellular and mouse experiments. J.Z., S.O. and T.M. performed zebrafish experiments. J.K. and A.M. performed intestinal organoid experiments. M.M. performed MS/MS experiments and data analysis. Y.S performed TCGA database analysis. Y.F. and T.H. performed immunofluorescent experiments. H.T. and S.T. provided technical assistance. T.I. supervised J.Z., S.O. and T.M. B-K.K supervised J.K. and A.M. H.T supervised T.H. K.I.N. supervised M.M. Y.O. supervised Y.F. T.T. supervised Y.S., S.T. and the project as a whole. T.T. and B-K.K. wrote the manuscript together with T.I., S.H. and input from all other authors.

## Competing interests

The authors declare no competing interests.
