## [Peer Review File · Nature Communications]

Reviewers' comments:

Reviewer #1 (Remarks to the Author):

Tsukiyama and collaborators have analyzed how phosphorylation of the tumor suppressor RNF43 influences its function. Their results indicate that phosphorylation of RNF43 at conserved serines is indispensable for the inhibition of Wnt signaling, embryonic development and intestinal homeostasis. Finally, the authors explore how RNF43 phosphorylation status affects other signaling pathways, Ras and p53, involved in colon carcinogenesis. They conclude that RNF43 and KRAS mutations cooperate via a Wnt-Ras-p53 axis thereby triggering tumorigenesis. The manuscript addresses a novel aspect in the regulation of RNF43, which is of high relevance in the field due to its important tumor suppressor role in many gastrointestinal tumors. The authors have conducted numerous experiments to determine how phosphorylation regulates RNF43 function and how this is relevant considering the number of mutations found to disrupt the phosphorylation site. However, it still remains unclear whether endogenous RNF43 is phosphorylated and what is the biological relevance of this at the end.

Major points:

1. Overexpression vs Endogenous

To show that RNF43 phosphorylation is important for its function, the authors perform experiments using STF293 cells transfected with different RNF43 constructs. These experiments do not demonstrate whether (endogenous) RNF43 is phosphorylated "under normal cellular conditions" as they suggest in page 6, line 8. To address this issue, the authors should compare RNF43 wild type expressing cells with mutant cells lacking the serine domain involved in phosphorylation. For instance, they should introduce the mutations by CRISPR/Cas9 in a cell expressing wild type RNF43 such as HT-29 cells, this would allow analysis of RNF43 phosphorylation and its effects under normal cellular conditions.

An important question which arises in this context is: What is the phosphorylation status of RNF43 in normal cells. So is RNF43 constitutively phosphorylated in normal cells, such as intestinal stem cells, and thus constitutively "UB-On"? This should be analyzed in organoids.

2. Localization

A major weakness of the manuscript and data, which is also linked to the first point, is the issue of RNF43 localization. While several papers claim that RNF43 is localized in the cytoplasm and/or at the cell membrane, the majority of reports finds RNF43 in the nucleus, especially when endogenous RNF43 is analyzed. This raises doubts about the results obtained with cytoplasmic RNF43.

The authors indicate that mutants in the serine domain do not show differences in protein-protein interaction and subcellular localization of the protein. However, the subcellular localization that the authors show is not the localization of endogenous RNF43, and the IF pictures they show here may merely show an artefact due to the forced expression of a GFP-labeled construct. The localization shown in Supp 3d is neither cytoplasmic nor nuclear, but it shows mainly an accumulation at the ER, which is a typical phenomenon observed when using overexpression of GFP fusion. Of note, the authors refer to previous publications which showed RNF43 to be expressed in the nucleus of the cells (References 19 and 32 included in their manuscript: (19: Shinada et al. RNF43 interacts with NEDL1 and regulates p53-mediated transcription. *Biochem Biophys Res Commun* 2010; 32: Xie et al. Association of RNF43 with cell cycle proteins involved in p53 pathway. *Int J Clin Exp Pathol* 2015). Therefore, it is unclear what the authors consider to be the "normal" subcellular localization of RNF43. The same holds true for the localization of the R127P mutant that they use in the study. They claim that it is aberrantly localized in the ER (Supp Fig 9c), but this localization seems to be the same for the other overexpressed phosphorylation mutants they are using and shown in Suppl Figure 3d. How can the authors conclude that RNF43 is expressed at the right subcellular compartment? These findings must be confirmed by using an antibody which detects the endogenous protein (recently, a monoclonal antibody was published "8D6" which detects endogenous RNF43 and works in IF, see Neumeyer et al, *Carcinogenesis* 2018; doi: 10.1093/carcin/bgy152, e.g. Suppl. Fig.1). As said above, when done in cells expressing endogenous RNF43 (with the respective mutants introduced by CRISPR/Cas), this would strongly corroborate the biochemical data obtained with overexpression experiments.

In this context, the authors repeatedly speak of subcellular localization or even aberrant ER localization throughout the manuscript. Given the concerns raised above, this statement is wrong until this has been proven for endogenous RNF43 bearing the respective mutations. The same is true for the paragraph in the discussion (p 14 line 16-20).

Minor points:

The tumor xenograft model is puzzling. The Ras active Cle-H3 cells used here seem not to form xenografts unless expressing the 3SA dominant negative form. This is in contrast to the literature, where the cells were reported to form tumors in nude mice.

HCT116 cells are mutated for Kras (p.G13D; c.38G>A). This should be mentioned in the text, and the authors should put this in context and explain why these cells were used. They especially need to explain the proposed mechanistic interaction between ras, p53 and RNF43.

R127P: the mutation is within the extracellular protease (PA) domain and abolishes Wnt inhibition. How does these oncogenic mutations impair phosphorylation of RNF43? If RNF43 is normally phosphorylated, as the authors claim, why is RNF43 not phosphorylated in the mutants? The authors should check for RNF43 phosphorylation status of R127P. R127P is a point mutation before the phosphorylation site: they speculate about mislocalization of the protein, but this is not clear: Why would mislocalization impair phosphorylation? In line with the comment above, this was performed under overexpression conditions of the mutant.

In this context: The authors use two transactivating constructs in the manuscript, R127P and 3SD. In the literature, another transactivating mutation, H292R/H295R was published, which seems also devoid of Fzd ubiquitination activity. It would add much to the understanding of the biochemical properties of RNF43 if all of these three mutants were compared directly regarding phosphorylation status, Fzd ubiquitination and localization. Again, would introduction of the phosphomimetic 3SD into the H292R/H295R mutant also abolish its transactivating ("oncogenic") activity?

Figure 1 a: The authors use a mutant d366-478 which shows transactivation in TOP-Luc. This mutant would also lack the NLS located between 432 and 440. Could it be that the observed effect is to a change in localization as well?

In Supplementary Figure 8c, the authors show that RNF43 mt is correlated with worse prognosis. On the other hand, it is well known that in CRC and GC, frameshift mutations correlate with better prognosis, and RNF43 often bears fs mutations. Thus, the data in this graph should be re-analyzed by separating RNF43 fs mutations from other RNF43 mutations.

The authors repeatedly use the expression "oncogenic RNF43" (p10, line 31, p 39, line 12). What they mean is RNF43 bearing transactivating mutations. However, since they show that RNF43 bearing transactivating mutations cannot transform NIH3T3 cells, I would avoid the term oncogenic. This should be rephrased throughout the manuscript.

P 1, line 30: do the authors really decline no potential conflict of interest. If so they must declare their COI.

P 5 line 17: explain/introduce STF-Luc assays

P 10 line 24: HCT116 cells that stably express RNF43 please change to HCT116 cells that stably overexpress RNF43

P 10 line 30-33: In my eyes the results do not reveal mechanistic insight between RNF43 and p53. The effect shown speculative is not proven mechanistically.

Taken together, this is a meticulously performed and very comprehensive study with huge potential impact for the field. This makes it especially important to address the aspects of endogenous RNF43 function and localization.

Reviewer #2 (Remarks to the Author):

In the manuscript by Tsukiyama, the authors focus on the phospho-switch of RNF43-mediated degradation of wnt receptors. The first part of the manuscript describes the effects of phosphorylation of RNF43 mainly using the Super Top Flash assay and the expression of Frizzleds on the cell membrane. This data is convincing and very clean.

My main concern with the manuscript is that there is no direct evidence that RNF43, or any of the mutants, affects endogenous Wnt signaling. Instead it seems that RNF43 is affecting many signaling pathways, as the authors mention on line 29-30 on page 10.

To obtain more biological significance, they used the zebrafish model and mouse colonospheres. Below are my comments.

1) While *tbx6* is influenced by Wnt signaling, it is typically used as a mesodermal marker. Mesoderm is induced by Nodal and patterned by Wnts. At this stage of development (30-50% epiboly), it is well established that Wnts are involved in dorsal-ventral patterning of the axis as well as in controlling the size of the organizer. (papers from the Lekven and Solnica-Krezel labs). *Tbx6* is not considered a direct target of the Wnt pathway and so it is difficult to determine if the effect of RNF43 is due to alterations in Wnt signaling or to other signaling mechanisms such as Nodal.

2) The TCF/Lef reporter fish was also used for quantitative expression of *egfp*. They showed a dose dependent decrease in *egfp* expression in the presence of wild-type but not 3SA RNF43. Similar to the Super TOP flash, this is an artificial reporter, which may not be indicative of the effect on endogenous Wnt targets.

3) The effect on both Wnt- β -catenin and Wnt-PCP is conflicting and would suggest that RNF43 does not discriminate between the two pathways. This could be determined by seeing if RNF43 overexpression could rescue the effect of Wnt-PCP specific *Wnt11* overexpression or Wnt- β -catenin specific *Wnt8* overexpression. They have very different phenotypes, which would allow the authors to discriminate between pathways.

4) The overexpression phenotypes look very different from one another. The short tail phenotypes may be a Wnt loss of function phenotype, but perturbation of BMP or FGF can also generate this phenotype.

5) One day phenotypes may be informative, but by this stage compensatory mechanisms typically obscure the main issue. From these images it appears that RNF43 is having an effect on more than just Wnt signaling.

6) Quantitative analysis of gene expression of specific Wnt target genes (*axin2*, *nkd1*, *sp5*) at 30-50% epiboly, would provide a more accurate assessment of RNF43 activity on Wnt signaling.

7) Qualitative analysis of organizer genes *gsc* and *chd* at 50% epiboly by whole mount in situ hybridization would provide a more accurate assessment of RNF43 activity on endogenous Wnt signaling.

They also performed some phenotypic analysis in mouse organoid cultures. Here they looked at organoid survival and infer that overexpression of wt or 3SD RNF43 mutants are lethal. It is difficult to interpret anything from these findings beyond this. The suggestion that RNF43 functions as an additional regulatory layer of Wnt signaling in ISCs is an overstatement. Quantification of endogenous Wnt target genes in these organoids or localization of β -catenin would help to support their statement. As an aside, in supplementary Figure 4e, the images at day 34 between 3SA and 3SD look very similar, like there is significant death going on in both. However, the organoids appear to recover only in 3SA at day 65. There is no explanation or acknowledgment of this.

Subsequently, the authors determine the effect of RNF43 mutants on tumor growth and find that the 3SA mutants results in more tumor growth compared to mock injected in the presence of activated Ras signaling. Further, the 3SA mutant in combination with activated Ras resulted in significantly more tumor burden when compared to wild-type or activated 3SD RNF43. For these experiments they used stable expression of RNF43 in Cle-H3 cell lines. Again, while the data looks very clean there is no indication that this is Wnt specific.

In other experiments they overexpress the various RNF43 mutants and find that they all inhibit p53 dependent activation of p21 and Bax and finally that phosphomimetic substitution can rescue the ubiquitination of frizzled 5. However, they do not demonstrate that this actually inhibits Wnt signaling.

In summary, the molecular data is very clean and there are clear results in their phenotypic assays but there is no clear indication which pathways are being affected. While several previous reports clearly demonstrate that RNF43 is involved in regulation of frizzled receptors, my concern is that the effects seen in this manuscript with the respect to the phospho-switch describe a novel function of RNF43.

Terry Van Raay

Submission of revised manuscript

We are delighted to resubmit our revised manuscript. The reviewers disclosed several problems encountered during their reading of our original work and suggested several critical experiments that have significantly improved our manuscript. The revised manuscript contains three additional important conclusions:

1. All key RNF43 mutations characterised in the first submission were confirmed to induce the same effects when introduced into the endogenous *RNF43* gene in STF293 cells.
2. RNF43 mainly acts at the cell surface, although it exerts a marginal but clear suppressive effect on Wnt signalling in the nucleus.
3. RNF43 regulates Wnt and p53 signalling concurrently via distinctive mechanisms.

In the revised manuscript, we focused on the role of endogenous RNF43, using CRISPR/Cas9 genome-engineered models of STE293. We also focused on the roles of RNF43 in the plasma membrane and nucleus, as well as in the Wnt and p53 signalling pathways during morphogenesis and tumorigenesis.

With these changes and additions, we believe that we have addressed all the technical issues raised by the reviewers. The revised manuscript is significantly improved, and our initial conclusion is supported by even more data.

A response to reviewer comments is included below.

Reviewer #1 (Remarks to the Author):

Tsukiyama and collaborators have analysed how phosphorylation of the tumor suppressor RNF43 influences its function. Their results indicate that phosphorylation of RNF43 at conserved serines is indispensable for the inhibition of Wnt signaling, embryonic development and intestinal homeostasis. Finally, the authors explore how RNF43 phosphorylation status affects other signaling pathways, Ras and p53, involved in colon carcinogenesis. They conclude that RNF43 and KRAS mutations cooperate via a Wnt-Ras-p53 axis thereby triggering tumorigenesis.

The manuscript addresses a novel aspect in the regulation of RNF43, which is of high relevance in the field due to its important tumor suppressor role in many gastrointestinal tumours. The authors have conducted numerous experiments to determine how phosphorylation regulates RNF43 function and how this is relevant considering the number

of mutations found to disrupt the phosphorylation site. However, it still remains unclear whether endogenous RNF43 is phosphorylated and what is the biological relevance of this at the end.

We appreciate the correct summary of our findings as proposed by that Reviewer #1. One main critical point that has been raised is the relevance of our findings in endogenous RNF43. To address the reviewer's criticism, we have generated several RNF43 mutants in STF293 cells using CRISPR/Cas9 technology. Our data confirm the following points:

1. Both endogenously expressed and exogenously overexpressed RNF43 proteins exhibited similar functions.
2. Endogenous RNF43 is expressed at an extremely low level in STF293 cells, as confirmed by the introduction of an HA-tag. This observation suggests that some of the requested experiments would need to be performed in an overexpression system.

We believe that our revised manuscript provides sufficient evidence to support a role for RNF43 phosphorylation in Wnt signalling and tumorigenesis. Please see our point-by-point replies below.

Major points:

1. Overexpression vs Endogenous

To show that RNF43 phosphorylation is important for its function, the authors perform experiments using STF293 cells transfected with different RNF43 constructs. These experiments do not demonstrate whether (endogenous) RNF43 is phosphorylated "under normal cellular conditions" as they suggest in page 6, line 8. To address this issue, the authors should compare RNF43 wild type expressing cells with mutant cells lacking the serine domain involved in phosphorylation. For instance, they should introduce the mutations by CRISPR/Cas9 in a cell expressing wild type RNF43 such as HT-29 cells, this would allow analysis of RNF43 phosphorylation and its effects under normal cellular conditions.

An important question which arises in this context is: What is the phosphorylation status of RNF43 in normal cells. So is RNF43 constitutively phosphorylated in normal cells, such as intestinal stem cells, and thus constitutively "UB-On"? This should be analyzed in organoids.

Reviewer #1 suggested the validation of our findings in cells expressing endogenous RNF43. We considered this to be a strong suggestion with regard to our data set. According to the mutation profiles of cancer cells in the database, many RNF43-expressing cancer cell lines harbour classical cancer mutations in *APC*, *TP53* and *KRAS*. Therefore, we initially decided to identify a suitable experimental model cell line for subsequent experiments. Moreover, strong RNF43 expression might suggest aberrant Wnt signalling activity, as *RNF43* is a downstream target of the Wnt pathway (Hao et al. Nature 2012. doi:10.1038/nature11019, Takahashi et al., PLoS One 2014. doi:10.1371/journal.pone.0086582, Tsukiyama et al. 2015. doi:10.1128/MCB.00159-15). Therefore, we did not use HT-29 cells that harboured APC mutations and were speculated having high Wnt activity in our revision experiments. Instead, we used STF293 cells, which express functional endogenous RNF43. First, we established a HA epitope knock-in at the C-terminal of RNF43 and used an anti-HA antibody to validate the weak but intact expression of endogenous RNF43 protein in STF293 cells (Supplementary Fig. 4a–e and page 8, line 15–29).

In response to the comment by Reviewer #1, we then used CRISPR/Cas9 genome editing to introduce several mutations into the endogenous *RNF43* gene in STF293 cells, and thus generated novel RNF43 KO, RNF43 Δ phospho-switch (Δ PS), and RNF43 R127P knock-in models (Fig. 2f–g). Previously, we and others reported that R-spondin increased surface Fzd expression and greatly facilitated Wnt signalling in HEK293T and STF293 cells (Kazanskaya et al. Dev Cell 2004. doi:10.1016/j.devcel.2004.07.019, Hao et al. Nature 2012. doi:10.1038/nature11019, Tsukiyama et al. 2015. doi:10.1128/MCB.00159-15). In other words, these cells express functional RNF43 and/or ZNRF3 proteins and so react to R-spondin treatment. Similarly, our RNF43 KO STF293 cells exhibited increased Wnt signalling (Fig. 2f), which confirmed the presence of functional RNF43 activity in this cell line. Other mutant cell lines harbouring RNF43 (Δ PS) and RNF43 (R127P) also exhibited enhanced Wnt activity, consistent with our overexpression experiments (Supplementary Fig. 3j). These data confirm that endogenous RNF43 behaved as expected, based on our overexpression experiments (page 8, line 29–page 9, line 6). We also confirmed the phosphorylation of endogenous RNF43 protein by CK1 in STF293 (RNF43-HA KI) cells (Supplementary Fig. 2f and page 8, line 1–2). Using our RNF43 (Δ PS) STF293 cells, we validated the importance of the phospho-switch in the regulation of RNF43 (Supplementary Fig. 2f). Moreover, we previously demonstrated that endogenous RNF43 and ZNRF3 act as

critical suppressors of Wnt signalling in both intestinal crypts and organoids (Koo et al. Nature 2012. doi:10.1038/nature11308), and that exogenous RNF43 (3SD) expression impairs the growth of intestinal organoids (Fig. 3c–d, Supplementary Fig. 5c–e). We believe these results, when taken together, sufficiently prove our initial claim that the phospho-switch of RNF43 regulates the activities of both endogenous and exogenous RNF43 in STF293 and intestinal organoids.

2. Localization

A major weakness of the manuscript and data, which is also linked to the first point, is the issue of RNF43 localization. While several papers claim that RNF43 is localized in the cytoplasm and/or at the cell membrane, the majority of reports finds RNF43 in the nucleus, especially when endogenous RNF43 is analysed. This raises doubts about the results obtained with cytoplasmic RNF43.

We agree with the point raised in this comment, but must disagree about the existence of clear evidence excluding the role of RNF43 at the cell membrane. We also emphasise that the existence of a role for RNF43 at the plasma membrane—namely the regulation of Fzd expression—does not exclude a possible nuclear function of RNF43. Although this point was not mentioned in our original submission, the lead author of this manuscript had previously described the nuclear localisation of RNF43 also in another publication (Tsukiyama et al. 2015. doi:10.1128/MCB.00159-15).

Several research groups have confirmed the regulatory role of RNF43 in terms of controlling the level of Fzd at the plasma membrane, using cross-disciplinary approaches ranging from cell biology, mouse genetics and structural biology. Most importantly, several reports described that RNF43 and ZNRF3 forms co-crystal structures with secreted and transmembrane proteins (Rspo-Lgr4/5/6), suggesting a role at the plasma membrane (Chen et al, Genes Dev 2013. doi: 10.1101/gad.219915.113, Zebisch et al. Nat Comm 2013. doi: 10.1038/ncomms3787). We have demonstrated that the RNF43 phospho-switch plays a critical regulatory role in the clearance of Fzd at the plasma membrane. Therefore, our current manuscript has focused mainly on this part of the RNF43 regulatory mechanism.

Nevertheless, in accordance with the reviewer's valuable comment, we assessed the role of the phospho-switch in the nuclear localisation and nucleus-associated function of RNF43. As noted, we confirmed the known subcellular localisation pattern of RNF43, which is

characterised by abundant expression in the endosome and at the nuclear periphery (Supplementary Fig. 4d). We also observed mild but clear suppressive RNF43 activity against nuclear β -catenin-driven Wnt signalling by Δ N- β -catenin (Supplementary Fig. 4h and page 9, line 23–30) as previously reported by the reviewer's group. Interestingly, none of our phospho-switch related mutants, RNF43 (3SA or 3SD), exhibited a discernible change relative to WT RNF43. In the same context, plasma membrane-driven Wnt activation via stimulation with the Wnt3a ligand was affected significantly by phospho-switch related mutants (Supplementary Fig 4h).

Taken together, our data suggest that the phospho-switch of RNF43 plays a critical role in the membrane-associated functions of RNF43 but has no discernible effect to the nuclear function of this protein.

The authors indicate that mutants in the serine domain do not show differences in protein-protein interaction and subcellular localization of the protein. However, the subcellular localization that the authors show is not the localization of endogenous RNF43, and the IF pictures they show here may merely show an artefact due to the forced expression of a GFP-labeled construct. The localization shown in Supp 3d is neither cytoplasmic nor nuclear, but it shows mainly an accumulation at the ER, which is a typical phenomenon observed when using overexpression of GFP fusion. Of note, the authors refer to previous publications which showed RNF43 to be expressed in the nucleus of the cells (References 19 and 32 included in their manuscript: (19: Shinada et al. RNF43 interacts with NEDL1 and regulates p53-mediated transcription. *Biochem Biophys Res Commun* 2010; 32: Xie et al. Association of RNF43 with cell cycle proteins involved in p53 pathway. *Int J Clin Exp Pathol* 2015).

In response to this comment from Reviewer #1, we examined the localisation of RNF43 using both EGFP-tagged and HA-tagged proteins. As noted, we used HA-tagging and IF staining to confirm the previously identified subcellular localisation pattern of RNF43, namely abundant expression in the endosome and at the nuclear periphery (Supplementary Fig. 4d).

As mentioned above, the current manuscript focused on the role of the RNF43 phospho-switch. Consistent with previous observations (Tsukiyama et al. 2015. doi:10.1128/MCB.00159-15), we observed perinuclear staining (Supplementary Fig. 4d)

and a mild inhibitory effect (~30%) on nuclear β -catenin in RNF43 overexpressing cells (Supplementary Fig. 4h), consistent with the results of a previous report (Loregger et al. *Sci Signal* 2015. doi: 10.1126/scisignal.aac6757). However, we did not observe differences in the abilities of various RNF43 phospho-mutants to suppress nuclear β -catenin and p53 activity (Fig. 5a, Supplementary Fig. 8 and page 13, line 13–17), suggesting that this phosphorylation event is dispensable for the nuclear activity of RNF43. The same mutant set had a significant effect on membrane-associated Wnt activation in response to Wnt3a ligand, as described above. Based on this data set, we again argue that our finding has no major impact on the nuclear role of RNF43.

Therefore, it is unclear what the authors consider to be the “normal” subcellular localization of RNF43. The same holds true for the localization of the R127P mutant that they use in the study. They claim that it is aberrantly localized in the ER (Supp Fig 9c), but this localization seems to be the same for the other overexpressed phosphorylation mutants they are using and shown in Suppl Figure 3d. How can the authors conclude that RNF43 is expressed at the right subcellular compartment? These findings must be confirmed by using an antibody which detects the endogenous protein (recently, a monoclonal antibody was published “8D6” which detects endogenous RNF43 and works in IF, see Neumeyer et al, *Carcinogenesis* 2018; doi: 10.1093/carcin/bgy152, e.g. Suppl. Fig.1). As said above, when done in cells expressing endogenous RNF43 (with the respective mutants introduced by CRISPR/Cas), this would strongly corroborate the biochemical data obtained with overexpression experiments.

Reviewer #1 has repeatedly suggested using the 8D6 monoclonal antibody to address the issue of endogenous RNF43 localisation. However, we did not employ this approach because of the following reasons:

1. Anti-RNF43(8D6) antibodies are not commercially available and
2. We could not determine the validity of this clone, given the very limited pool of reported information (Neumeyer et al. *Carcinogenesis* 2018. doi; 10.1093/carcin/bgy152).

Instead, we employed CRISPR/Cas9 gene editing technology to generate a HA epitope knock-in in STF293 cells, which allowed us to utilise highly specific anti-HA antibodies. First, we inserted a nucleotide sequence encoding the HA epitope at the C-terminal end of the endogenous *RNF43* gene (Supplementary Fig. 3a–e). The introduction of this tag did not

induce any discernible changes to Wnt3a-induced signalling activation, suggesting that endogenous RNF43-HA was fully functional (Supplementary Fig. 3d). Although the endogenous RNF43 protein was not detectable via simple IB and IF experiments, it could be detected after enrichment by immunoprecipitation (Supplementary Fig. 3c,3e), suggesting an extremely low level of endogenous expression in STF293 cells. Taken together with the above-described results from STF293-RNF43 KO or Δ PS cells (Supplementary Fig. 3f-j), we conclude that STF293 cells express a small amount of functional RNF43 under a basal and biological level of Wnt/ β -catenin signalling activity (page 8, line 15–page 9, line 6). This conclusion was also supported by our previous identification of the rapid turnover of RNF43 protein ($t_{1/2}$ = 30–40 min) (Tsukiyama et al. 2015. doi:10.1128/MCB.00159-15).

In this context, the authors repeatedly speak of subcellular localization or even aberrant ER localization throughout the manuscript. Given the concerns raised above, this statement is wrong until this has been proven for endogenous RNF43 bearing the respective mutations. The same is true for the paragraph in the discussion (p 14 line 16-20).

Our results suggest that RNF43 acts mainly to suppress Wnt signalling by downregulating Fzd. In this manuscript, we demonstrated that the RNF43 phospho-switch regulates the membrane function but not the nuclear function of RNF43. As noted by the reviewer, we observed a clear ability of RNF43 to suppress nuclear β -catenin. However, the phospho-switch mutant did not appear to affect the nuclear activity of RNF43. Again, we emphasise that the nuclear function of RNF43 was not the main focus of our study, as the phosphorylation status had little effect on this particular mechanism of RNF43. In contrast, phosphorylation had a very clear effect on the membrane function of RNF43 (Supplementary Fig. 4h and page 9, line 23–30).

Minor points:

The tumor xenograft model is puzzling. The Ras active Cle-H3 cells used here seem not to form xenografts unless expressing the 3SA dominant negative form. This is in contrast to the literature, where the cells were reported to form tumours in nude mice.

The original paper (Takiguchi et al. Clin Exp Metastasis 1992. PMID: 1505125) demonstrated that tumorigenic activity of Cle-H3 cells varied according to the number of in

vitro passages. Moreover, this tumorigenic activity could be recovered via in vivo cell growth and recloning. We believe that our Cle-H3 mock cells had a lower level of the tumorigenic activity than the originally described cells. Nevertheless, this weaker tumorigenic activity allowed us to evaluate the oncogenic role of mutant RNF43 in our model. We confirm that we used the same cell passage in our analysis. Moreover, we were able to observe various tumorigenic activities of our Cle-H3 cells, specifically in colony formation (Fig. 4b, 6b, Supplementary Fig. 7b), spheroid culture (Fig. 4c) and in vivo tumour formation (~10% of injected mice) (Fig. 4a, 4d, 6c Supplementary Fig. 7c). Therefore, we believe that our results remain reliable and demonstrate the differential oncogenic activities associated with various *RNF43* mutations.

HCT116 cells are mutated for Kras (p.G13D; c.38G>A). This should be mentioned in the text, and the authors should put this in context and explain why these cells were used. They especially need to explain the proposed mechanistic interaction between ras, p53 and RNF43.

We appreciate the reviewer's comment. In response, we have included this description in our manuscript (page 13, lines 10–11). HCT116 cells carry a frameshift mutation in *RNF43* and wild-type p53. Therefore, we could examine the functional effect of exogenously expressed RNF43 on p53 signalling without accommodating endogenous RNF43 activity.

R127P: the mutation is within the extracellular protease (PA) domain and abolishes Wnt inhibition. How does these oncogenic mutations impair phosphorylation of RNF43? If RNF43 is normally phosphorylated, as the authors claim, why is RNF43 not phosphorylated in the mutants? The authors should check for RNF43 phosphorylation status of R127P. R127P is a point mutation before the phosphorylation site: they speculate about mislocalization of the protein, but this is not clear: Why would mislocalization impair phosphorylation?

The mechanism by which mislocalisation impairs phosphorylation remains unclear. However, we determined a lower level of phosphorylation and reduced functional activity with RNF43 (R127P) than with WT RNF43 (Supplementary Fig. 2i, Fig. 6d and page 8, line 6–10). The introduction of phospho-mimetic mutations to RNF43 (R127P-3SD) restored the

suppressive activity of this protein, as well as its ability to ubiquitinylate Fzd. However, the 3SD substitution did not change the localization of R127P mutant (Supplementary Fig. 10c). These results suggest that RNF43 activity is regulated by phosphorylation associated with subcellular localisation (i.e., phosphorylation at an appropriate localisation between the Golgi and cell surface), rather than the subcellular localisation itself (Fig. 2h).

Accordingly, we propose the following order of molecular mechanisms: 1) correct localisation, 2) phosphorylation by CK1, 3) ubiquitination of Fzd, and 4) endocytosis and lysosomal degradation of Fzd.

In this context: The authors use two transactivating constructs in the manuscript, R127P and 3SD. In the literature, another transactivating mutation, H292R/H295R was published, which seems also devoid of Fzd ubiquitination activity. It would add much to the understanding of the biochemical properties of RNF43 if all of these three mutants were compared directly regarding phosphorylation status, Fzd ubiquitination and localization. Again, would introduction of the phosphomimetic 3SD into the H292R/H295R mutant also abolish its transactivating (“oncogenic”) activity?

We tested the effect of a phosphomimetic 3SD mutation on RNF43 (H292R) and did not observe any functional rescue (Supplementary Fig. 10a and page14, line 32–page 15, line 2). This result was somewhat expected, as the H292R mutation directly impairs the function of the RING-finger domain. Consequently, RNF43 (H292R) would not possess ubiquitination activity even in the presence of phosphomimetics. As explained above, we believe that the molecular mechanism occurs in the following order: 1) proper localisation, 2) phosphorylation of CK1, 3) ubiquitination of Fzd, and 4) endocytosis and lysosomal degradation of Fzd.

We also showed that another PA-domain mutation (I48T) could be rescued by introducing phosphomimetic mutations, thus further supporting our proposed mechanism (Supplementary Fig. 10a).

Figure 1 a: The authors use a mutant d366-478 which shows transactivation in TOP-Luc. This mutant would also lack the NLS located between 432 and 440. Could it be that the observed effect is to a change in localization as well?

We demonstrated the transactivation activity of the mutant $\Delta 366\text{--}478$ in Fig. 1a, as noted by Reviewer #1. We also tested many other RNF43 deletion mutants in our previous report (Tsukiyama et al. Mol Cell Biol 2015. doi: 10.1128/MCB.00159-15), where we observed that several mutants lacking a potential NLS (432–440); m1 ($\Delta 366\text{--}783$), m4 ($\Delta 366\text{--}442$), m10 ($\Delta 318\text{--}595$) and m11 ($\Delta 318\text{--}783$) retained the ability to attenuate Wnt signalling. We also observed the transactivation of $\Delta SRR2$ ($\Delta 465\text{--}479$) with the intact NLS and RING domain (Supplementary Fig. 1f). If the reviewer's assumption were correct, we would observe the opposite result. Therefore, we conclude that the loss of NLS did not have a major effect on the ability of RNF43 to suppress Wnt signalling.

In Supplementary Figure 8c, the authors show that RNF43 mt is correlated with worse prognosis. On the other hand, it is well known that in CRC and GC, frameshift mutations correlate with better prognosis, and RNF43 often bears fs mutations. Thus, the data in this graph should be re-analyzed by separating RNF43 fs mutations from other RNF43 mutations.

We note that we had discussed this point in the previous version of our manuscript. The RNF43(G659fs) mutation is mainly observed in microsatellite instability (MSI) CRC, which is associated with a better prognosis (page 18, lines 6–10) (Giannakis et al. Nat Genet 2014. doi:10.1038/ng.3127, Bond et al. Oncotarget 2016. doi:10.18632/oncotarget.12130). We agree with the reviewer's suggestion to eliminate MSI CRC from the survival rate analysis. We have accordingly included the prognostic outcomes without MSI CRC patients (Supplementary Fig. 9c and page 14, line 3–5). When MSI CRC cases were excluded from the total sample of CRC cases, the prognoses of patients with and without MSI CRC were similar.

The authors repeatedly use the expression “oncogenic RNF43” (p10, line 31, p 39, line 12). What they mean is RNF43 bearing transactivating mutations. However, since they show that RNF43 bearing transactivating mutations cannot transform NIH3T3 cells, I would avoid the term oncogenic. This should be rephrased throughout the manuscript.

The word “oncogenic” can be defined both broadly and narrowly. In a broad sense, an oncogenic mutation simply refers to a mutation that can induce tumorigenesis (with/without

other factors). In a narrow sense, oncogenic mutation refers specifically to single mutation sufficient to induce tumorigenesis in the absence of other factors. As we have used the broad meaning of this term in our manuscript, we would like to retain this usage.

P 1, line 30: do the authors really decline no potential conflict of interest. If so they must declare their COI.

We agree with the reviewer and have made a declarative statement on page 2, line 6.

P 5 line 17: explain/introduce STF-Luc assays

This description has been added to page 6, lines 9–11.

P 10 line 24: HCT116 cells that stably express RNF43 please change to HCT116 cells that stably overexpress RNF43.

This has been corrected on page 13, line 11.

P 10 line 30-33: In my eyes the results do not reveal mechanistic insight between RNF43 and p53. The effect shown speculative is not proven mechanistically.

We and others previously reported that RNF43 suppresses p53 pathway activity (Nailwal et al. Cell Death Dis 2015. doi:10.1038/cddis.2015.131; Fig. 5 and Fig. S8 in our manuscript) in a dose-dependent manner (Shinada et al. BBRC 2011. doi:10.1016/j.bbrc.2010.11.082). However, we did not examine the activity of the p53 pathway in tumour samples harbouring RNF43 mutants in our xenograft experiments.

Therefore, we have toned down our conclusion from "yield" to "suggest," as per the reviewer's suggestion (page 13, line 20).

Taken together, this is a meticulously performed and very comprehensive study with huge potential impact for the field. This makes it especially important to address the aspects of endogenous RNF43 function and localization.

We appreciate Reviewer 1's enthusiasm regarding the improvements to our manuscript. We hope that our revision has addressed all questions and concerns and now meets the standard for publication in *Nature Communications*.

Reviewer #2 (Remarks to the Author):

In the manuscript by Tsukiyama, the authors focus on the phospho-switch of RNF43-mediated degradation of Wnt receptors. The first part of the manuscript describes the effects of phosphorylation of RNF43 mainly using the Super Top Flash assay and the expression of Frizzleds on the cell membrane. This data is convincing and very clean.

My main concern with the manuscript is that there is no direct evidence that RNF43, or any of the mutants, affects endogenous Wnt signaling. Instead it seems that RNF43 is affecting many signaling pathways, as the authors mention on line 29-30 on page 10.

We appreciate the comments and suggestions made by Reviewer #2 in terms of defining the role of RNF43 in endogenous Wnt signalling more clearly, especially during zebrafish development. Below, we have provided new datasets related to this point.

To obtain more biological significance, they used the zebrafish model and mouse colonospheres. Below are my comments.

1) While *tbx6* is influenced by Wnt signaling, it is typically used as a mesodermal marker. Mesoderm is induced by Nodal and patterned by Wnts. At this stage of development (30-50% epiboly), it is well established that Wnts are involved in dorsal-ventral patterning of the axis as well as in controlling the size of the organizer. (papers from the Lekven and Solnica-Krezel labs). *Tbx6* is not considered a direct target of the Wnt pathway and so it is difficult to determine if the effect of RNF43 is due to alterations in Wnt signaling or to other signaling mechanisms such as Nodal.

Reviewer #2 suggested that *Tbx6* may not be a suitable reporter gene for determining Wnt signalling activity. First, we agree that other factors can modulate *Tbx6* expression; for example, BMP and Wnt activate the *Tbx6* promoter via two different regions (Szeto and Kimelman. *Development* 2004. doi: 10.1242/dev.01236). The reviewer argued that other signalling pathways, such as BMP or Nodal, could have influenced *Tbx6* expression in our experiments when 30–50% of fish embryos were in the epiboly stage. Nevertheless, we emphasise first that the *Tbx6* promoter is regulated directly by Wnt signalling, and many previous reports have used *Tbx6* expression as an endogenous reporter of Wnt activity (He et al. *Cell Discov* 2017. doi: 10.1038/celldisc.2017.3, Hino et al. *Dev Biol* 2017. doi: 10.1016/j.ydbio.2017.11.016, Shimizu et al. *Cell Rep* 2014. doi: 10.1016/j.celrep.2014.07.040, Li et al. *PNAS* 2011. doi: 10.1073/pnas.1009353108, Ding et al. *J Cell Biol* 2008. doi: 10.1083/jcb.200803147, Rui et al. *Dev Cell* 2007. doi:10.1016/j.devcel.2007.07.006). Furthermore, we reintroduced previous findings regarding the role of RNF43. Specifically, multiple groups have reported a specific role of RNF43 in the Wnt pathway, with minimal or no effects on other pathways such as Notch, TNF, FGFR and Eph (Tsukiyama et al. *Mol Cell Biol* 2015. doi: 10.1128/MCB.00159-15, Koo et al. *Nature* 2012. doi:10.1038/nature11308).

However, we also agree with Reviewer #2 that a more reliable marker of endogenous Wnt activity is needed. To this end, we used qPCR to examine the expression of the known direct downstream targets of Wnt signalling, *Axin2* (Yan et al. *PNAS* 2001. doi: 10.1073/pnas.261574498, Lustig et al. *Mol Cell Biol* 2002. doi: 10.1128/mcb.22.4.1184-1193.2002, Jho et al. *Mol Cell Biol* 2002. doi: 10.1128/mcb.22.4.1172-1183.2002) and *Nkd1* (Zeng et al. *Nature* 2000. doi: 10.1038/35001615, Van Raay et al. *Dev Biol* 2007. doi:10.1016/j.ydbio.2007.04.018, Larraguibel et al. *Mol Biol Cell* 2015. doi: 10.1091/mbc.E14-12-1648). WT-RNF43 suppressed the expression of all endogenous Wnt target genes and an artificial Wnt reporter gene, whereas this expression was enhanced by the RNF43-3SA mutant in a dose-dependent manner. Concurrently, RNF43 did not perturb the expression of *Sef* and *Id1*, which are well-established target genes of FGF and BMP signalling, respectively (Fig. 3b and page 10, line 22–25) (Tsang et al. *Nat Cell Biol* 2002. Doi: 10.1038/ncb749, Fürthauer et al. *Nat Cell Biol* 2002. doi: 10.1038/ncb750, Warga et al. *Dev Biol* 2013. doi:10.1016/j.ydbio.2013.08.018, Neugebauer et al. *Development* 2013.

doi:10.1242/dev.096388, Das et al. Dev Cell 2019. doi: 10.1016/j.devcel.2019.05.014, Korchynskiy et al. J Biol Chem 2002. doi: 10.1074/jbc.M1111023200, Pouget et al. Nat Comm 2014. doi: 10.1038/ncomms6588, Thorimbert et al. FASEB J 2015. doi: 10.1096/fj.15-272955, Das et al. Dev Cell 2019. doi: 10.1016/j.devcel.2019.05.014). Taken together, these data suggest that RNF43 acts as a negative regulator in the embryonic Wnt signalling pathway but had very little or no influence on the other examined pathways.

2) The TCF/Lef reporter fish was also used for quantitative expression of egfp. They showed a dose dependent decrease in egfp expression in the presence of wild-type but not 3SA RNF43. Similar to the Super TOP flash, this is an artificial reporter, which may not be indicative of the effect on endogenous Wnt targets.

As suggested, we examined different endogenous markers, such as *Axin2*, *Nkd1*, and *Tbx6*, in embryos at different stages (45–50% and 75–80% epiboly). All our results demonstrated a strong correlation between these endogenous markers and the TCF/LEF-eGFP reporter signal (Fig. 3b and Supplementary Fig. 5a and page 10 line 18–21).

3) The effect on both Wnt- β -catenin and Wnt-PCP is conflicting and would suggest that RNF43 does not discriminate between the two pathways. This could be determined by seeing if RNF43 overexpression could rescue the effect of Wnt-PCP specific Wnt11 overexpression or Wnt- β -catenin specific Wnt8 overexpression. They have very different phenotypes, which would allow the authors to discriminate between pathways.

We and others reported previously that the expression of WT-RNF43/ZNRF3 induces a short A-P axis in the developing zebrafish embryo (Supplementary Fig. 5b and Hao et al. Nature 2012. doi:10.1038/nature11019) or xenopus embryo (Tsukiyama et al. Mol Cell Biol 2015. doi: 10.1128/MCB.00159-15). The short A-P axis phenotype is induced by both canonical and noncanonical Wnt signalling. A loss of canonical Wnt signalling induces axial truncation due to mesodermal depletion (Takada et al. Genes Dev 1994. doi:10.1101/gad.8.2.174, Yoshikawa et al. Dev Biol 1997. doi:10.1006/dbio.1997.8502), while the loss of noncanonical Wnt signalling induces axial shortening due to reduced PCP and CE movement (Yamaguchi et al. Development 1999. PMID:10021340, Veeman et al.

Dev Cell 2003. doi:10.1016/S1534-5807(03)00266-1). Both canonical and noncanonical Wnt signalling requires Fzd receptor activity.

In our previous studies, we demonstrated that RNF43 could promote the degradation of various Fzd receptors involved in both canonical and noncanonical Wnt signalling (Koo et al. Nature 2012. doi:10.1038/nature11308, Tsukiyama et al. Mol Cell Biol 2015. doi: 10.1128/MCB.00159-15). Accordingly, we thought that the RNF43 phospho-switch might regulate Fzd receptors in both signalling pathways. In developing zebrafish embryos, we demonstrated that the phospho-switch could regulate not only canonical Wnt target genes (*Axin2*, *Nkd1*, *Tbx6*) but also the noncanonical Wnt-related phenotype, namely the “short & wide” axis phenotype (Fig. 3a–b, Supplementary Fig. 5a–c). As this point has been well explored in previous reports (Hao et al. Nature 2012. doi:10.1038/nature11019), we believe that our simple analysis is sufficient to support our conclusion.

4) The overexpression phenotypes look very different from one another. The short tail phenotypes may be a Wnt loss of function phenotype, but perturbation of BMP or FGF can also generate this phenotype.

Our explanation of our results might have been unclear. Overexpression of WT RNF43 induced a considerable reduction in Wnt/ β -catenin signalling activity that mimicked the Wnt-loss phenotype (short or no tail), whereas the opposite phenomenon was observed with overexpression of the phospho-switch mutant RNF43-3SA (Supplementary Fig. 5c). These observations are consistent with other data obtained from our cell line experiments, wherein that the phospho-switch of RNF43 was shown to regulate Wnt signalling (Fig. 1, 2, Supplementary Fig. 1–3).

Several groups have studied the role of RNF43 in the Wnt pathway (Hao et al. Nature 2012. doi:10.1038/nature11019, Koo et al. Nature 2012. doi:10.1038/nature11308, de Lau et al. Dev Cell 2014. doi:10.1101/gad.235473.113, Tsukiyama et al. Mol Cell Biol 2015. doi:10.1128/MCB.00159-15, Hao et al. Cancers 2016. doi:10.3390/cancers8060054, Nusse and Clevers. Cell 2017. doi:10.1016/j.cell.2017.05.016) including Reviewer #1 (Loregger et al. Sci Signal 2015. doi: 10.1126/scisignal.aac6757). Therefore, we do not consider it necessary to re-confirm this point in every new publication. In accordance with the Reviewer #2's comment, however, we evaluated the levels of phosphorylated Smad, p38 and ERK and thus determined that RNF43 did not affect BMP and FGF signalling (Fig. R1). In other

words, RNF43 overexpression had no discernible effect on the downstream phosphorylation of components in the BMP and FGF signalling pathways. We also confirmed that RNF43 overexpression did not perturb the endogenous target genes of the FGF (*Sei*) and BMP (*Id1*) signalling pathways in zebrafish embryos at the 45–50% epiboly stage (Fig. 3b and page 10, line 18–25).

Fig. R1, RNF43 does not function in BMP and FGF signalling

5) One day phenotypes may be informative, but by this stage compensatory mechanisms typically obscure the main issue. From these images it appears that RNF43 is having an affect on more than just Wnt signaling.

As mentioned above, the role of RNF43 in the Wnt signalling pathway has been well established in several of papers, including some published by Reviewer #1 (Hao et al. Nature 2012. doi:10.1038/nature11019, Koo et al. Nature 2012. doi:10.1038/nature11308, Tsukiyama et al. Mol Cell Biol 2015. doi:10.1128/MCB.00159-15, Hao et al. Cancers 2016. doi:10.3390/cancers8060054, Nusse and Clevers Cell 2017. doi:10.1016/j.cell.2017.05.016, Loregger et al. Sci Signal 2015. doi: 10.1126/scisignal.aac6757). Although we cannot fully exclude the possibility that RNF43 affects other pathways, we feel that this point has already been addressed sufficiently and is not the main focus of our study. Rather, we aimed to describe the important roles of the RNF43 phospho-switch in the suppression of Wnt pathway signalling.

To clarify this matter for Reviewer #2, we provide the following brief explanation of our previous and current work. We demonstrated that RNF43 targets Wnt but not TNF, Notch (Tsukiyama et al. Mol Cell Biol 2015. doi: 10.1128/MCB.00159-15), FGF or BMP signalling (Fig. R1, Fig. 3b) signalling. Although RNF43 overexpression reduced the surface levels of Fzd and Lrp5/6, other receptors, such as Notch, EGFR and Eph (Koo et al. Nature 2012. doi: 10.1038/nature11308), were not affected.

6) Quantitative analysis of gene expression of specific Wnt target genes (*axin2*, *nkd1*, *sp5*) at 30-50% epiboly, would provide a more accurate assessment of RNF43 activity on Wnt signaling.

We agree with this comment from Reviewer #2. Accordingly, we also examined the expression of *axin2*, *nkd1* and *tbx6* as specific endogenous target genes of Wnt, as well as an artificial Wnt reporter gene, *Egfp*, in embryos at 45–50% epiboly (Fig. 3b). We also evaluated *Tbx6* and *Egfp* at 75–80% epiboly (Supplementary Fig. 5a). In all these experiments, we observed that RNF43 suppressed both endogenous and exogenous Wnt target genes in a dose-dependent manner (page 10, line 18–22). In contrast, the RNF43-3SA mutant enhanced the expression of these genes, suggesting that the functions of the RNF43 phospho-switch in developing zebrafish embryos and cultured cells are similar.

7) Qualitative analysis of organizer genes *gsc* and *chd* at 50% epiboly by whole mount in situ hybridization would provide a more accurate assessment of RNF43 activity on endogenous Wnt signaling. They also performed some phenotypic analysis in mouse organoid cultures. Here they looked at organoid survival and infer that overexpression of wt or 3SD RNF43 mutants are lethal. It is difficult to interpret anything from these findings beyond this. The suggestion that RNF43 functions as an additional regulatory layer of Wnt signaling in ISCs is an overstatement. Quantification of endogenous Wnt target genes in these organoids or localization of β -catenin would help to support their statement. As an aside, in supplementary Figure 4e, the images at day 34 between 3SA and 3SD look very similar, like there is significant death going on in both. However, the organoids appear to recover only in 3SA at day 65. There is no explanation or acknowledgment of this. Subsequently, the authors determine the effect of RNF43 mutants on tumor growth and find

that the 3SA mutants results in more tumor growth compared to mock injected in the presence of activated Ras signaling. Further, the 3SA mutant in combination with activated Ras resulted in significantly more tumor burden when compared to wild-type or activated 3SD RNF43. For these experiments they used stable expression of RNF43 in Cle-H3 cell lines. Again, while the data looks very clean there is no indication that this is Wnt specific. In other experiments they overexpress the various RNF43 mutants and find that they all inhibit p53 dependent activation of p21 and Bax and finally that phosphomimetic substitution can rescue the ubiquitination of frizzled 5. However, they do not demonstrate that this actually inhibits Wnt signaling.

This question is very lengthy. However, we recognise the main point regarding the strong doubt of Reviewer #2 for the role of RNF43 in the Wnt signalling pathway. This doubt is rather surprising, as this role of RNF43 has been very well established in the field as mentioned above (Hao et al. Nature 2012. doi:10.1038/nature11019, Koo et al. Nature 2012. doi:10.1038/nature11308, Tsukiyama et al. Mol Cell Biol 2015. doi:10.1128/MCB.00159-15, Hao et al. Cancers 2016. doi:10.3390/cancers8060054, Loregger et al. Sci Signal 2015. doi: 10.1126/scisignal.aac6757, de Lau et al. Genes Dev 2014. doi:10.1101/gad.235473.113, Nusse and Clevers. Cell 2017. doi:10.1016/j.cell.2017.05.016). We believe that the new sets of data included in the revised version of the manuscript will resolve all of the reviewer's concerns. However, we also emphasise that with this study, we aimed to evaluate the role of RNF43 phospho-switch with respect to the known functions of RNF43, using previously developed assays that measured the surface level of Fzd, activity of Wnt signalling, development of zebrafish embryos, growth of intestinal cells, tumorigenesis and other aspects. We aimed mainly to compare the differences between RNF43 WT/3SD and 3SA and to understand the importance of the newly identified RNF43 phospho-switch, for which very consistent data have been derived across multiple systems (e.g., human and mouse cell lines, zebrafish morphogenesis and mouse organoids).

In summary, the molecular data is very clean and there are clear results in their phenotypic assays but there is no clear indication which pathways are being affected. While several previous reports clearly demonstrate that RNF43 is involved in regulation of frizzled receptors, my concern is that the effects seen in this manuscript with the respect to the phospho-switch describe a novel function of RNF43.

We agree with the reviewer that previous reports clearly demonstrated the regulation of Fzd receptors by RNF43. In this study, we tested whether our newly identified phospho-switch plays a significant role in this RNF43-mediated regulatory process. As per our results and newly added data, we have determined that the phospho-switch specifically regulates the ubiquitination and endo-lysosomal degradation of Fzd (Fig. 1–2, 6d, Supplementary Fig. 1–3, 4i) but has little or no effect on the RNF43-mediated suppression of nuclear β -catenin and p53 (Fig. 5a, Supplementary Fig. 4h, 8). These findings suggest that the phospho-switch is highly specific for a single pathway and single mechanism. During this revision process, we tested other pathways (e.g., FGF, BMP) and mechanisms (e.g., RNF43-mediated suppression of nuclear β -catenin and p53) and did not find evidence of phospho-switch involvement. We have discussed these new results in the revised manuscript.

Terry Van Raay

Reviewers' comments:

Reviewer #1 (Remarks to the Author):

The authors have addressed most of the comments. However, the phosphorylation status of RNF43 under endogenous conditions remains unresolved. The cellular model used to explore phosphorylation of RNF43 is not adequate, since STF293 cells barely express RNF43. Therefore, the statement "All key RNF43 mutations characterised in the first submission were confirmed to induce the same effects when introduced into the endogenous RNF43 gene in STF293 cells" cannot be made. I do not understand why the authors chose a cell line which barely expresses RNF43 to corroborate their findings. There are several cell lines which do express wt RNF43.

The authors argue that "Endogenous RNF43 is expressed at an extremely low level in STF293 cells, as confirmed by the introduction of an HA-tag. This observation suggests that some of the requested experiments would need to be performed in an overexpression system." I do not agree. This observation should have prompted them to choose another cell line.

I would like to point out that I have no intention of being petty or belaboring the issue, but there are so many findings out there which rely exclusively on overexpression and turn out to not hold true under endogenous conditions that I find it extremely important that such biochemical findings are corroborated under endogenous conditions.

I do not agree with the explanation of the authors that HT29 cells would not be a suitable model because of constitutive Wnt signaling. Several papers have shown that RNF43 can still exert its function also in the presence of constitutive Wnt signaling.

In addition, authors have not addressed RNF43 phosphorylation in non-transformed cells (organoids), which is essential to conclude their claims under endogenous conditions.

Regarding the issue of localization, I did not mean to question the fact that RNF43 can be found at the cell membrane and interaction with Fzd, this is well documented. I wanted to indicate that results obtained by overexpression may be misleading, since the overexpressed constructs in most cases seem to show non-physiological localization. This is especially obvious in the Supplementary Fig. 4d mentioned by the authors, where a EGFP-RNF43 fusion protein is mostly found in the endosome/ER. This localization is highly artificial and therefore not helpful to substantiate the findings. I do recommend to completely remove Supplementary Fig. 4d. Rather, I would like to see the localization of endogenous wt and RNF43 phospho-mutants. The authors introduced an HA tag into RNF43 in STF293, but failed to detect this tagged protein due to low expression levels. Thus, this point could not be clearly addressed. I still recommend to perform investigate localization of endogenous wt and RNF43 phospho-mutants in a suitable cellular context. I fully understood that "the nuclear function of RNF43 was not the main focus of our study", as the authors state, but in the light of the fact that especially in the Wnt pathway, phosphorylation status can be an important regulator of subcellular localization, I would find it important that this aspect is unambiguously clarified.

Reviewer #2 (Remarks to the Author):

The revised manuscript by Tsukiyama et al incorporates substantially more data than the first submission as noted in their response. They have addressed my concern regarding observing the effect of RNF43 and the 3SA mutant on endogenous Wnt targets (Figure 3; Suppl Fig 5). A new concern I have is their use of replicates in generating the statistics. Throughout the manuscript the authors routinely use the term "N=3 biological replicates". However, for the qPCR data in Figure 3b, they use 30 embryos and it appears that here their ANOVA is based on one biological replicate using a pool of 30 embryos. In Suppl. Fig 5, they state n=3 technical replicates and base their ANOVA on these technical replicates. In contrast to their other data, there is no indication that they used biological replicates. This may be a typo or an omission, but given they are seeing modest (but statistically significant) changes in gene expression (2X decrease, ~1.7X increase) I think it is very important to show biological replicates.

A final major concern I have is the argument the authors make regarding the role of RNF43 as a cell membrane bound antagonist of Wnt signaling. The lack of membrane localization of RNF43 to the membrane is a major shortcoming in the manuscript, especially when there are other publications demonstrating its localization to both the plasma and nuclear membranes and their own data shows no membrane localization. How do the authors reconcile the function of RNF43 at the membrane while at the same time showing no membrane localization of the protein? The authors make the argument that RNF43 targets the Fzd receptor for ubiquitination and lysosomal degradation. I don't argue this fact and the data are pretty clear, but contrast with their imaging results. Also, they provide no evidence that this is occurring on the plasma membrane or that it is a direct mechanism. Indeed, they clearly show that membrane localization is not required for Fzd ubiquitination as the R127P mutant is mislocalized, yet the R127P 3SD mutant is sufficient to ubiquitinate Fzds. Their previous publication used fractionation assays to show mostly membrane enrichment in western blots, but showed no images. They state that others have demonstrated membrane localization via a physical interaction by co-IPs and crystal structure so it seems likely. Indeed, it may be that the 2 dimensional nature of cells in culture preclude an apical-basal polarity (that exists in crypt cells), for proper localization of RNF43. However, they have mature organoids in culture overexpressing RNF43. Organoids have been shown to have apical-basal polarity. The authors need to look at the distribution of RNF43 in their organoid cultures. This would go a long way to validating their model. I recognize that many papers make the statement that RNF43 is a membrane bound E3 ubiquitin ligase, but only a few show its subcellular localization, which has been shown to be both nuclear and plasma membrane localized.

Given the variability over where RNF43 functions (nucleus, nuclear membrane, plasma membrane), the number of papers describing its many functions (p53, Wnt/PCP; Wnt/ β -catenin; NEDL1; E-cadherin), it is only prudent that its localization be addressed fully and convincingly for this manuscript to be published in a higher end journal.

Terry Van Raay

Point-by-point responses to Reviewers' comments (shown in blue):

Reviewer #1 (Remarks to the Author):

The authors have addressed most of the comments. However, the phosphorylation status of RNF43 under endogenous conditions remains unresolved.

In response to the comments from reviewer #1, we established lines of RNF43-HA knock-in STF293 cells and showed that functional **endogenous** RNF43 is expressed in STF293 cells (Fig. 2g, Sfig. 3e, 3f, 3l). In addition, **endogenous RNF43 is naturally phosphorylated** by CK1 (Sfig. 2f) similar to overexpressed RNF43 in *in vitro* kinase assays (Sfig. 2e, 2g). Furthermore, the R127P mutation that induces ER localisation of endogenous RNF43 had reduced phosphorylation compared to WT (Sfig. 2i, 2j). The details of each of these are discussed below.

Sfig. 3e showed that HA-tagged **endogenous** RNF43 is expressing in STF293 cells.

Sfig. 3f showed that HA-tagging of **endogenous** RNF43 does not change the function of endogenous RNF43.

[REDACTED]

Fig. 2g (upper left) and Sfig. 3l (upper right) showed respectively that HA-tagged endogenous RNF43 mutants upregulate surface Fzd expression and accelerate Wnt signalling independently of Rspo similar to when these components are overexpressed in wildtype STF293 cells. [Fig. 1b (middle left), 1d (middle right) and Fig. 5D (lower) of our previous report; Tsukiyama et al. 2015. doi:10.1128/MCB.00159-15].

Sfig. 2e, 2f, 2g showed that **endogenous** RNF43 is **naturally** phosphorylated by CK1 (upper left) similar to the overexpression of RNF43 (upper right and lower).

i

Sfig. 2i and 2j showed that R127P mutation which induces ER localisation (upper) greatly declined **endogenous** RNF43 phosphorylation (lower) in STF293 cells.

The cellular model used to explore phosphorylation of RNF43 is not adequate, since STF293 cells barely express RNF43. Therefore, the statement “All key RNF43 mutations characterised in the first submission were confirmed to induce the same effects when introduced into the endogenous RNF43 gene in STF293 cells” cannot be made. I do not understand why the authors chose a cell line which barely expresses RNF43 to corroborate their findings. There are several cell lines which do express wt RNF43.

The authors argue that “Endogenous RNF43 is expressed at an extremely low level in STF293 cells, as confirmed by the introduction of an HA-tag. This observation suggests that some of the requested experiments would need to be performed in an overexpression system.” I do not agree. This observation should have prompted them to choose another cell line.

If the low level of RNF43 expression is a reason to disregard all data obtained from STF293 cells, then we also have to ignore all knock-out or knockdown experiments performed on zebrafish and mice with RNF43 and ZNRF3 in the past, where their level of expression is also low. The choice of cell line was made based on two important aspects. 1) It should have functional RNF43 and 2) it also has a normal Wnt signalling pathway. We believe that the STF293 cells included in our study sufficiently covers these points. This is because these cells have functional endogenous RNF43. Upon Wnt stimulation, the level of surface Fzd was down-regulated and these cells do not have any known mutation in the canonical Wnt signalling pathway.

In comparison, the suggested cell lines with excessive RNF43 expression carry Wnt-activating genomic mutations. As previously reported, RNF43 is a direct target of the Wnt pathway. Indeed, cells with abundant RNF43, which reviewer #1 pointed out, show hyper Wnt activation due to mutations in APC and/or β -catenin (HT-29, APC; Caco-2, APC; AGS, APC and CTNNB1). These critical mutations that induce uncontrollable β -catenin accumulation are at the downstream of Wnt ligands, Fzd receptors and RNF43, which we are examining in this study. Due to the aberrant downstream activation of Wnt signalling, it is very

difficult to assess the function of RNF43 in general in these suggested cancer cell lines as illustrated in detail below.

Difference between STF293 and RNF43 abundant cancer cells

STF293 cells weakly express RNF43 protein due to basal level of Wnt signal and its rapid turnover ($t_{1/2} \sim 0.5h$). Wnt stimulation leads accumulation of β-catenin ① and expression of Wnt target genes including RNF43 ②. RNF43 suppresses Wnt signals via degradation of Fzd ③ and capture of Tcf4 ④, and establishes a negative feedback loop. This is a natural mechanism in Wnt signal regulation with RNF43. In this healthy context, we can detect bidirectional change of signalling activity in whole Wnt pathway ⑤.

Meanwhile in RNF43 abundant cancer cells, LOF mutation of APC leads an abnormal accumulation of β-catenin ⑥ and the expression of Wnt target RNF43 ⑦. These abundant RNF43 protein may localize on the membrane at nuclear and/or at other cellular compartments. Surface RNF43 may downregulate Fzd ⑧ but we cannot detect the biological change of Wnt activity, because saturated level of β-catenin is already expressing at the downstream of surface event due to APC mutation ⑥. These cells are insensitive for the event on cell surface ⑨ (independent to ligands; Wnt and Rspo as in Sfig. 3a). On the other hand, RNF43 on nuclear membrane can still capture Tcf4 and suppress Wnt signal ⑩ (Sfig. 4h). Therefore, these cells are sensitive only for the nuclear event which is at the downstream of β-catenin accumulation ⑪.

For this discussion, we used data from a previous publication (Farin et al. Nature 2016. doi:10.1038/nature16937) that clearly showed **the effect of endogenous RNF43/ZNRF3 in the regulation of membrane-bound Fzd level**, as shown below. In APC knock-out organoids, due to the elevation of RNF43/ZNRF3, the membrane level of Wnt3-labelled Fzd is cleared out (3a, right, white arrows), which was then restored by blocking the RNF43/ZNRF3 function by Rspo (3a, left). In case of RNF43/ZNRF3 double

knock-out organoids, there was no downregulation of the surface level of Wnt3-Fzd (3b). These previous data obtained from primary intestinal organoids are highly similar to our mechanistic model on how RNF43/ZNRF3 works in STF293 cells (3e), except that we have a novel finding on the function of RNF43/ZNRF3 phosphorylation.

[REDACTED]

Nevertheless, we functionally confirmed again that the STF293 cells we used in this study express endogenous RNF43 on the cell surface in order to support our statement. Similar to primary intestinal organoids (Farin et al. Nature 2016. doi:10.1038/nature16937), the following were observed: 1) Rspo facilitates Wnt ligand-induced signal activation by increasing the surface level of Fzd (Sfig. 3a, 3b, shown below); 2) The activation of Wnt/ β -catenin signalling by CHIR treatment suppressed surface Fzd expression via negative feedback circuit as reported by many studies (Sfig. 3b); and 3) The suppression of Fzd expression via RNF43 induction was impaired by Rspo (Sfig. 3b).

These results suggest that STF293 cells retain an intact Wnt signalling cascade. In contrast to this, the suggested CRC cells that express abundant levels of endogenous WT RNF43 due to the saturated accumulation of β -catenin by APC mutation (e.g. HT-29 cells) failed to show ligand-induced signal activation (Sfig. 3a). Moreover, Rspo also fail to facilitate the accumulation of active β -catenin in HT-29 cells, unlike in STF293 cells (Sfig. 3a), suggesting that active β -catenin is aberrantly saturated and the functionality of Rspo-Lgr4/5-RNF43/ZNRF3-Fzd axis was completely disorganised in these CRC cells (described in page 8 of revised manuscript, shown in red), as illustrated above. Thus, we concluded that the STF293 cell line is a better qualified model for examining the endogenous regulation of Wnt signalling and the role of RNF43/ZNRF3 phosphorylation (which is the main aim of our study) rather than other CRC cells with APC or β -catenin mutations, even though STF293 cells express lower level of endogenous RNF43 compared to CRC cells.

I would like to point out that I have no intention of being petty or belaboring the issue, but there are so many findings out there which rely exclusively on overexpression and turn out to not hold true under endogenous conditions that I find it extremely important that such biochemical findings are corroborated under endogenous conditions.

We agree with the reviewer that assessing the endogenous function of a protein is very important. In line with the reviewer's comment, we and others (Koo et al. Nature 2012. doi:10.1038/nature11308, Farin et al. Nature 2016. doi:10.1038/nature16937 shown before, Koo PNAS 2015. doi/10.1073/pnas.1508113112, shown below) have provided enough data supporting the function of endogenous RNF43 in regulating Wnt signalling

via the surface level of Fzd, as discussed above. Primary intestinal organoids lacking R/Z do not require Rspo for their expansion (B–C vs E–F, below).

[REDACTED]

I do not agree with the explanation of the authors that HT29 cells would not be a suitable model because of constitutive Wnt signaling. Several papers have shown that RNF43 can still exert its function also in the presence of constitutive Wnt signaling.

We would like to consider the reports that the reviewer mentioned here. We hope that the reviewer can kindly relay to us the information found in those reports. As explained above, the function of endogenous RNF43 in regulating the surface level of Fzd is clearly present in both intestinal organoids (Farin et al. Nature 2016. doi:10.1038/nature16937) and in 293 cells, which have been reported by several papers and by multiple groups. The phospho-regulation only affects the RNF43 function in the membrane, and not in the nucleus (Sfig. 4h). Moreover, Wnt-Rspo-induced signal activation (Sfig. 3b) cannot be detected in HT-29 cells and, therefore, cannot be used for testing our hypothesis.

In addition, authors have not addressed RNF43 phosphorylation in non-transformed cells (organoids), which is essential to conclude their claims under endogenous conditions.

We feel that the experiments suggested by reviewer #1 are nearly impractical. To examine it with current materials and techniques, we have to newly generate RNF43-HA knock-in mouse and isolate the intestinal stem cells for organoid culture. Using these materials, we should label endogenous RNF43 with ³²P in

organoid, and then immunoprecipitate endogenous RNF43 using anti-HA antibodies for autoradiography analysis to detect endogenous phosphorylation. The reviewer would hopefully agree that this will take too much time. As the editor wanted to know, we believe that it is impractical to conduct this experiment at this point. It also will not add much data, as the endogenous phospho event has already been well analysed (Fig. 2f, 2g, Sfig. 2f, 2i).

Regarding the issue of localization, I did not mean to question the fact that RNF43 can be found at the cell membrane and interaction with Fzd, this is well documented. I wanted to indicate that results obtained by overexpression may be misleading, since the overexpressed constructs in most cases seem to show non-physiological localization.

This is especially obvious in the Supplementary Fig. 4d mentioned by the authors, where a EGFP-RNF43 fusion protein is mostly found in the endosome/ER. This localization is highly artificial and therefore not helpful to substantiate the findings. I do recommend to completely remove Supplementary Fig. 4d. Rather, I would like to see the localization of endogenous wt and RNF43 phospho-mutants.

The authors introduced an HA tag into RNF43 in STF293 but failed to detect this tagged protein due to low expression levels. Thus, this point could not be clearly addressed. I still recommend to perform investigate localization of endogenous wt and RNF43 phospho-mutants in a suitable cellular context. I fully understood that “the nuclear function of RNF43 was not the main focus of our study”, as the authors state, but in the light of the fact that especially in the Wnt pathway, phosphorylation status can be an important regulator of subcellular localization, I would find it important that this aspect is unambiguously clarified.

We are happy to see that the reviewer admitted that the main function of RNF43 in the cell membrane is to regulate the surface level of Fzd. Unlike the claims of the reviewer, this mechanism has been confirmed with endogenous RNF43 in both organoids and in STF293 cells (Fig. 2g, Farin et al. Nature 2016. doi:10.1038/nature16937, shown above). The overall observation is clearly in line with data obtained by overexpressed RNF43.

The essential point that reviewer #1 is concerned about is the localisation of endogenous RNF43 protein with and without the R127P oncogenic mutation. We recently obtained additional data that endogenous RNF43 (both WT and R127P mutant) similarly localises in heavy microsome membrane fractions, and only the R127P mutant was localised in the ER membrane in significant amounts, although WT is negligible. This is not immunofluorescent staining data as the reviewer wanted to see, but still gives us solid evidence supporting the presence of oncogenic RNF43 stacks in the ER at the endogenous level (Sfig 2j, shown below again).

Supplementary Fig. 2j, RNF43(R127P) mutant translocates to ER. Endogenous RNF43-HA WT and R127P mutant expressed similarly both in the nuclear and heavy microsome (HM) fractions. Only R127P mutant showed the ER localization in this HM fraction, whereas WT does not.

Reviewer #2 (Remarks to the Author):

The revised manuscript by Tsukiyama et al incorporates substantially more data than the first submission as noted in their response. They have addressed my concern regarding observing the effect of TNF43 and the 3SA mutant on endogenous Wnt targets (Figure 3; Suppl Fig 5). A new concern I have is their use of replicates in generating the statistics. Throughout the manuscript the authors routinely use the term “N=3 biological replicates”. However, for the qPCR data in Figure 3b, they use 30 embryos and it appears that here their ANOVA is based on one biological replicate using a pool of 30 embryos. In Suppl. Fig 5, they state n=3 technical replicates and base their ANOVA on these technical replicates. In contrast to their other data, there is no indication that they used biological replicates. This may be a typo or an omission, but given they are seeing modest (but statistically significant) changes in gene expression (2X decrease, ~1.7X increase) I think it is very important to show biological replicates.

We apologise for the error in the previous version of our manuscript. They should have been ‘n = 3, technical replicates with pool of 30 (Fig. 3b) or 32–38 (Sfig. 5a) embryos’. We have corrected the description in the figure legend (Sfig. 5b). Furthermore, we strongly agree with the reviewer’s suggestion that having biological replicates is important. Therefore, we performed additional experiments repeatedly in order to obtain biological replicates and make our results more reliable. We have replaced Fig. 3b and corrected the description in figure legend as ‘n = 3, biological replicates with pools of 25-30 embryos’.

A final major concern I have is the argument the authors make regarding the role of RNF43 as a cell membrane bound antagonist of Wnt signaling. The lack of membrane localization of RNF43 to the membrane is a major shortcoming in the manuscript, especially when there are other publications demonstrating its localization to both the plasma and nuclear membranes and their own data shows no membrane localization. How do the authors reconcile the function of RNF43 at the membrane while at the same time showing no membrane localization of the protein? The authors make the argument that TNF43 targets the Fzd receptor for ubiquitination and lysosomal degradation. I don’t argue this fact and the data are pretty clear, but contrast with their imaging results. Also, they provide no evidence that this is occurring on the plasma membrane or that it is a direct mechanism.

First of all, the idea that a protein would show plasma membrane localisation due to its role in the membrane is not correct in all instances. There are many examples showing the membrane recruitment of a protein upon

ligand-receptor engagement and receptor phosphorylation. The static image only shows the major place of localisation of a protein. Indeed, another group showed that Fzd localises in the intracellular compartment in the absence of Rspo treatment in organoid cultures, although Fzd is a Wnt receptor which should be functioning on the cell surface (Farin et al. Nature 2016. doi:10.1038/nature16937, shown below).

Our previous report, along with the report of others (Hao et al. Nature 2012. doi:10.1038/nature11019, Koo et al. Nature 2012. doi:10.1038/nature11308, Tsukiyama et al. 2015. doi:10.1128/MCB.00159-15) showed that RNF43/ZNRF3 ubiquitinates Fzd and induces the endocytosis and lysosomal degradation of Fzd.

[REDACTED]

Moreover, we also previously showed that RNF43/ZNRF3 DKO organoids do not require Rspo for Wnt-dependent growth, although it is essential for WT organoids (Koo et al. PNAS 2015. doi/10.1073/pnas.1508113112, shown below, B–C vs E–F).

[REDACTED]

These endogenous results in organoid culture are completely in line with our observations made in STF293 cells in this manuscript (Sfig. 3l, shown below), strongly supporting our conclusion in this manuscript.

Indeed, they clearly show that membrane localization is not required for Fzd ubiquitination as the R127P mutant is mislocalized, yet the R127P 3SD mutant is sufficient to ubiquitinate Fzds.

We have never stated that membrane localisation is not required for RNF43 function. We provided additional data (Sfig. 2j, shown below) to support the idea that an RNF43-R127P mutant is present in the heavy

microsome and is stacked on the ER membrane, as shown in our previous report with overexpressed mutants (Tsukiyama et al. 2015. doi:10.1128/MCB.00159-15). Mimicking forced phosphorylation of serines recovers the function of RNF43-R127P to ubiquitinate Fzd (Fig. 6d), downregulates Wnt signalling (Fig. 6a) and suppresses tumorigenesis (Fig. 6b, 6c), even if it retains abnormal localisation (Sfig. 10c).

Their previous publication used fractionation assays to show mostly membrane enrichment in western blots, but showed no images. They state that others have demonstrated membrane localization via a physical interaction by co-IPs and crystal structure so it seems likely. Indeed, it may be that the 2 dimensional nature of cells in culture preclude an apical-basal polarity (that exists in crypt cells), for proper localization of RNF43. However, they have mature organoids in culture overexpressing RNF43. Organoids have been shown to have apical-basal polarity. The authors need to look at the distribution of RNF43 in their organoid cultures.

We appreciate this point. It has been extremely difficult to visualise the endogenous expression of RNF43 using valid antibodies. Therefore, we took the HA tag knock-in approach in the previous revision. Although we failed in the immunostaining of endogenous RNF43, we have recently obtained clear biochemical data that most endogenous RNF43 is present in the HM membrane (Sfig. 2j, shown above). Moreover, we are happy to discuss data from a previous publication from another laboratory that clearly shows the effect of RNF43/ZNRF3 in the regulation of membrane-bound Wnt3-Fzd level in organoids (Farin et al. Nature 2016. doi:10.1038/nature16937, shown below again). In APC knock-out organoids, membrane level of Wnt3-Fzd is cleared out due to the elevation of RNF43/ZNRF3, which is then restored by blocking the RNF43/ZNRF3 function by Rspo. In case of RNF43/ZNRF3 double knock-out organoids, there is no downregulation of the surface level of Wnt3-Fzd. These results were obtained from primary intestinal organoids that reviewer #2 suggested to use, and are in line with our results using STF293 cells with both endogenous and overexpressed RNF43 proteins.

[REDACTED]

This would go a long way to validating their model. I recognize that many papers make the statement that RNF43 is a membrane bound E3 ubiquitin ligase, but only a few show its subcellular localization, which has been shown to be both nuclear and plasma membrane localized. Given the variability over where RNF43 functions (nucleus, nuclear membrane, plasma membrane), the number of papers describing its many functions (p53, Wnt/PCP; Wnt/ β -catenin; NEDL1; E-cadherin), it is only prudent that its localization be addressed fully and convincingly for this manuscript to be published in a higher end journal.

We also believe that E3s can have many targets with multiple roles, similar to kinases that have many substrates. Indeed, other ligases, Skp2, β TrCP and Fbx7, ubiquitinate multiple substrates (Frescas et al. *Nat Rev Cancer* 2008. doi:10.1038/nrc2396, Yeh et al. *Mol Cancer* 2018. doi.org/10.1186/s12943-018-0857-2). RNF43/ZNRF3 may have similar characteristics. However, this does not change their main role in the Wnt signalling pathway by promoting ubiquitination-mediated endo-lysosomal degradation of Fzd. Given the fact that RNF43/ZNRF3 are recognised as crucial negative feedback regulators of the Wnt pathway by many

researchers in the field, the novel regulatory mechanism by RNF43 phosphorylation would be of general interest.

Terry Van Raay

Reviewers' comments:

Reviewer #3 (Remarks to the Author):

The submitted manuscript from Tsukiyama et al. describe a novel aspect of RNF43 regulation, revealing a conserved C-terminal serine-triplicate site that undergoes obligate phosphorylation via CK1 for the ubiquitination of Fzd receptors, and subsequent inhibition of WNT signalling. The authors carefully delineate the requirement of RNF43-phosphorylation through the generation of multiple phospho-mimetic and phospho-dead RNF43 constructs to unpick their effect on WNT signalling and Fzd turnover. They use these constructs to assess the functional requirement of RNF43-phosphorylation in tissue development, homeostasis and tumourigenesis. The authors finally show that restoration of RNF43-phosphorylation in mutant RNF43 cancer cells is sufficient to reduce WNT signalling via Fzd degradation, which is sufficient for reduced cancer cell growth and offer this as a potential therapeutic modality for WNT-ligand sensitive cancers. However, this approach is limited to a very small cohort of patients with mutations in the relevant phosphorylation sites. While the molecular characterisation demonstrated in STF293 and NIH3T3 cells is of high quality, subsequent functional characterisations are tested in less than ideal model systems. Additionally, the manuscript has several points that are difficult to reconcile with previous studies and require explanation/further testing.

Points for revision:

1. Previous work from co-authors (Koo et al. 2012, Nature) demonstrate that loss of both Rnf43 and Znr3 is required to induce aberrant WNT signalling and tissue hyperplasia, and loss of Rnf43 or Znr3 alone is insufficient to induce appreciable changes to tissue homeostasis or WNT signalling. In support, Neumeyer et al. also show that Rnf43 only induces changes in the stomach mucosa and not the intestine. In contrast, your data suggest single Rnf43 mutants are sufficient to induce tumour formation compared to WT. This is difficult to reconcile. It would be very helpful to assay the expression of Znr3 in most, if not all, of the functional assays described in the manuscript.
2. Figure 2b. Does S478D/E reduce Fzd surface expression? This may inform about the sequence of phosphorylation.
3. Figure 3a-b and Supp. Figure 5c. The authors claim RNF43-phosphorylation regulates both 'canonical' and 'non-canonical' WNT-regulated aspects of zebrafish gastrulation. To formally test the regulation of non-canonical WNT, injection of various RNF43 constructs should be performed in b-catenin ko animals.
4. Figure 3c-d, Supp. Figure d-f. The authors infer growth changes to organoid cultures relate to changes in ISC maintenance. If so, please show associated changes to WNT signalling, ISCs and differentiated cell types in the various conditions. Also, if RNF43-3SA organoid cultures are RSPO-independent, do they lose their wild-type morphology and change to cystic? Pictures of viral-transduced organoids are shown to be cystic, and it is known that preparation for viral transduction forces this morphological change (CHIR, ROCKi, WNT3a etc.). The use of intestinal organoids are an excellent surrogate for intestinal biology. However, you should refrain from stating the observations made in organoids relate to intestinal homeostasis, this is not accurate.
5. Figure 4a. These experiments should be performed using virally-transduced intestinal organoids, which are much more robust and appropriate than NIH3T3 cells. This would also support or contrast the findings described by Neumeyer et al. 2018.
6. Figure 4b-d. Given the authors highlight the co-occurrence of RNF43 and KRAS mutation in pancreatic cancer, pancreatic cancer cell lines mutant for Kras (AsPC1, DANG, Capan1) should be used instead of Cle-H3 cells.

7. Do the RNF43-3SA tumour xenografts display changes in WNT signalling/high nuclear β -catenin? This should be examined.

8. The link between p53 and phosphorylation of RNF43 is not convincing, and relies on the expression of p21 and Bax following DNA damage. WNT is sufficient to repress

p21 via Myc in both p53 WT and mutant CRC cells (van de Wetering, 2002, Cell). Shouldn't you see some changes in p21 in your RNF43 mutant conditions, given they alter WNT signalling? If phosphorylation of RNF43 regulates aspects of p53 signalling, but not expression, would it be worthwhile testing RNF43 mutant constructs in p53-mutant CRC cells?

9. With respect to co-operation between Ras and RNF43 in CRC, a more appropriate mutation combination is Braf and RNF43. Perhaps, the authors could test the efficacy of their RNF43 mutant constructs in organoids from Braf-mutant mice. This approach would better reflect the serrated CRC subtype.

Additional point

As the authors show that the RNF43-HA tag acts in a similar manner to endogenous RNF43. I am comfortable with the use of this reagent, endogenous RNF43 is not so easy to work with.

Point-by-point response to Reviewers' comments (shown in blue):

Reviewer #3 (Remarks to the Author):

The submitted manuscript from Tsukiyama et al. describe a novel aspect of RNF43 regulation, revealing a conserved C-terminal serine-triplicate site that undergoes obligate phosphorylation via CK1 for the ubiquitination of Fzd receptors, and subsequent inhibition of WNT signalling. The authors carefully delineate the requirement of RNF43-phosphorylation through the generation of multiple phospho-mimetic and phospho-dead RNF43 constructs to unpick their effect on WNT signalling and Fzd turnover. They use these constructs to assess the functional requirement of RNF43-phosphorylation in tissue development, homeostasis and tumorigenesis. The authors finally show that restoration of RNF43-phosphorylation in mutant RNF43 cancer cells is sufficient to reduce WNT signalling via Fzd degradation, which is sufficient for reduced cancer cell growth and offer this as a potential therapeutic modality for WNT-ligand sensitive cancers. However, this approach is limited to a very small cohort of

patients with mutations in the relevant phosphorylation sites. While the molecular characterisation demonstrated in STF293 and NIH3T3 cells is of high quality, subsequent functional characterisations are tested in less than ideal model systems. Additionally, the manuscript has several points that are difficult to reconcile with previous studies and require explanation/further testing.

We appreciate your contribution as the third reviewer of our manuscript. We also thank for your constructive comments and suggestions. We carefully read all the comments and found that some points raised by the reviewer are either caused by misunderstanding due to insufficient explanation of our data or by some unresolved conflicts in the field, in case of which we must say that it is out of our main scope – **the phospho-regulation of RNF43**.

In response to the comments from the reviewer, we performed a number of new experiments to support our main conclusion, '**phospho-switch of RNF43 controls Wnt signalling and tumorigenesis**', and revised our manuscript accordingly. Here, we submit the fourth version of our manuscript, which has been revised for two years. We hope that this revision answers all the

concerns of the reviewer with regard to the importance of phospho-switch function of RNF43.

Points for revision:

1. Previous work from co-authors (Koo et al. 2012, Nature) demonstrate that loss of both Rnf43 and Znr3 is required to induce aberrant WNT signalling and tissue hyperplasia, and loss of Rnf43 or Znr3 alone is insufficient to induce appreciable changes to tissue homeostasis or WNT signalling. In support, Neumeyer et al. also show that Rnf43 only induces changes in the stomach mucosa and not the intestine. In contrast, your data suggest single Rnf43 mutants are sufficient to induce tumour formation compared to WT. This is difficult to reconcile. It would be very helpful to assay the expression of Znr3 in most, if not all, of the functional assays described in the manuscript.

We agreed with the comment from the reviewer that the absence of both RNF43 and ZNRF3 is required to upregulate Wnt signalling and to induce hyperplasia in the intestinal tissue. Hence, considering the expression of ZNRF3 is important as suggested. However, we would like to draw the attention of the reviewer to the dominant negative nature of RNF43 missense mutations. Previous studies have demonstrated that missense mutants (e.g. RING mutant) of RNF43 have an aberrant Wnt stimulatory activity even in the presence of ZNRF3 (Hao et al. Nature 2012, DOI: 10.1038/nature11019, Koo et al. Nature 2012, DOI: 10.1038/nature11308, Tsukiyama et al. MCB 2015. DOI: 10.1128/MCB.00159-15). Therefore, our phosphor-defective missense mutation in RNF43 promotes Wnt activity even in the presence of ZNRF3. In this sense, most missense mutants of RNF43 are different from null mutants, which require concomitant deletion of ZNRF3 to exhibit a similar phenomenon. We believed that this dominant negative function of RNF43 missense mutants is a very well-known knowledge among scientists studying RNF43, and it is not surprising as the reviewer pointed out. We also believe that it is totally consistent with previous reports from our group and others.

Nevertheless, in response to this comment, we examined the expression of ZNRF3 with or without WT and mutant RNF43. The expression level of ZNRF3 was not perturbed by all RNF43 proteins examined in both STF293 cells and zebrafish embryos (page 9, lane 30–31, Fig. S4e, shown below).

Also, we rephrased ‘loss of β -catenin-mediated inhibition’ into ‘gain of β -catenin-mediated facilitation’ in order to emphasise the dominant negative effect of RNF43 mutant (page 13, line 8). Furthermore, we have illustrated in Figure 4a and S7b (page 13, line 5–9) that only the dominant negative mutation of RNF43 is insufficient for tumorigenesis unless coupled with KRAS activation. These results would answer the reviewer’s question.

2. Figure 2b. Does S478D/E reduce Fzd surface expression? This may inform about the sequence of phosphorylation.

Please see Figure 2d. We already showed in our manuscript that RNF43-WT, RNF43-3SD and RNF43-S478D downregulate the surface Fzd expression in a normal state. The inhibition of CK1 by IC261 treatment restores Fzd levels with WT and S478D mutants, but not with 3SD (Fig. 2d). We also confirmed that the phosphorylation of S478 is functionally upstream of P-S474–476 (3S) as shown in Figure 2c. These results suggest that S478 is a priming phosphorylation for sequential 3S phosphorylation by CK1. In response to the reviewer’s comment, we revised the word ‘increased’ into ‘restored’ in order to clearly describe the function of S478D (page 7, line 30).

3. Figure 3a-b and Supp. Figure 5c. The authors claim RNF43-phosphorylation regulates both 'canonical' and 'non-canonical' WNT-regulated aspects of zebrafish gastrulation. To formally test the regulation of non-canonical WNT, injection of various RNF43 constructs should be performed in β -catenin ko animals.

We agree with the concern that elongation of body axis is a coordinated result of Wnt/ β -catenin signalling that provides mesodermal cells and noncanonical Wnt signalling which rearranges cells in a convergent extension movement. However, we have already demonstrated the involvement of RNF43 in noncanonical Wnt signalling (Tsukiyama et al. MCB 2015, DOI: 10.1128/MCB.00159-15). Similarly, we have also illustrated in this report that RNF43 functions not only in the canonical Wnt/ β -catenin signalling (Fzd downregulation) but also in the noncanonical Wnt signalling (Fzd downregulation and Dvl). We also demonstrated that RNF43 mutants that cannot bind to or ubiquitinate Fzd still retain the ability to suppress the elongation of animal cap (Tsukiyama et al. MCB 2015. DOI: 10.1128/MCB.00159-15). These results suggest that noncanonical Wnt signalling suppresses axis elongation by RNF43 in *Xenopus laevis* embryos despite the accumulation of Fzd receptors.

In our opinion, evaluating the anterior-posterior axis elongation in the absence of β -catenin can be very complicated as it does not correctly reflect the activity of noncanonical Wnt signalling, due to the depletion of gastrulation-derived mesodermal cells necessary for somitogenesis in axis elongation (Takada et al. Genes Dev 1994, DOI: 10.1101/gad.8.2.174, Yoshikawa et al. Dev Biol 1997, DOI: 10.1006/dbio.1997.8502). Due to this reason, such experiment has never been performed before. Moreover, loss of β -catenin itself can further disrupt normal morphogenesis by compromising the dorsoventral axis formation in the development of zebrafish. The function of molecules in noncanonical Wnt signalling has been discussed in the previous study based on the changes in notochord thickness and PAM

width in zebrafish embryos (Shimizu et al. Cell Rep 2014, DOI: 10.1016/j.celrep.2014.07.040).

Taken together, in our opinion, we believe that it is unfeasible to correctly examine the function of RNF43 and its mutants in noncanonical Wnt signalling by using β -catenin KO animals. Nevertheless, in response to the reviewer's comment, we revised our description for the role of RNF43 phospho-switch in noncanonical Wnt signalling (page 11, lane 21).

4. Figure 3c-d, Supp. Figure d-f. The authors infer growth changes to organoid cultures relate to changes in ISC maintenance. If so, please show associated changes to WNT signalling, ISCs and differentiated cell types in the various conditions. Also, if RNF43-3SA organoid cultures are Rspo-independent, do they lose their wild-type morphology and change to cystic? Pictures of viral-transduced organoids are shown to be cystic, and it is known that preparation for viral transduction forces this morphological change (CHIR, ROCKi, WNT3a etc.). The use of intestinal organoids are an excellent surrogate for intestinal biology. However, you should refrain from stating the observations made in organoids relate to intestinal homeostasis, this is not accurate.

We would like to apologise for causing the misunderstanding that made the reviewer ask these questions due to our insufficient explanation. Firstly, we would like to emphasise that RNF43-3SA phospho-mutant used in this study acts as a dominant negative mutant and does not autonomously activate Wnt signalling in a Wnt-independent fashion, despite that R519x mutant can weakly activate the signal without Wnt ligand as informally reported recently (Spit et al. bioRxiv 2020, DOI: 10.1101/748574). RNF43-3SA and RNF43- Δ PS mutants are dominant negative mutants that induce the accumulation of Fzd in a Rspo-independent manner (Figs. 2f, 2g, S1g, and S3l) and accelerates Wnt signalling still in a Wnt-dependent manner. Therefore, the expression of 3SA mutant in intestinal organoids does not lead cystic morphology in the absence of exogenous Wnt in culture condition (ENR; EGF, Noggin, R-spondin) (Fig. 3c and S5d–e) but develop cystic organoids in the presence of Wnt3a CM (WENR; Wnt + ENR) (Fig. 3d and S5f). Furthermore, we already indicated that RNF43 phospho-defective

mutant exhibits Rspo independency in the intestinal organoid model (Figs. 3c, d and S5d–f; page 10, line 31–page 11, line 6) when Rspo concentration was low (3SA, 10% vs 1%) as well as in cellular experiments (Fig. S3l; page 9, line 17–18), whereas the growth of 3SD mutant organoids retains a strong Rspo dependency (3SD, 10% vs 1%) in the same culture condition (Fig. S5e). A similar result was obtained from a different culture condition (WENR, with Wnt3a CM) (Fig. S5f), confirming the requirement of a strong Rspo stimulation to counteract the action of the RNF43-3SD mutant.

In order to prevent further misunderstanding, we additionally presented a graph indicating that RNF43-3SA or RNF43-3SD phospho-mutants respectively accelerate or suppress Wnt signalling in a Wnt-dependent manner as illustrated in Figure S1i (shown below). In the absence of Wnt ligand, RNF43 does not make any significant change in Wnt signalling (page 6, lane 33–page 7, lane 2).

It was difficult to assess the advantage and disadvantage of RNF43 phosphorylation in organoid growth in the previous version of the manuscript. To further support our results obtained from biochemical, cellular and allograft tumorigenesis experiments, we additionally presented the graph in Figure 3c which shows the advantage of RNF43-3SA mutant in the viability assay of organoids at day 4 post-RNF43 expression (shown below). Induction of RNF43-3SA by tamoxifen (4-OHT) increases the viability of organoid contrary to 3SD mutant that reduces it (page 12, line 3). Similar observation was also found in the result at day 3 (Fig. S5d). These

results are consistent with our other results obtained in this study. All results suggest the functional importance of the RNF43 phospho-switch.

Nevertheless, in response to the suggestion by Reviewer #3, we revised our explanation for intestinal homeostasis. Instead of the term 'homeostasis', we have made a correction of it as 'stem cell maintenance' (page 10, line 32 and page 11, line 30), which is fairly easy to assess using organoid culture system.

We hope that these of additional information could resolve the concerns of the reviewer.

5. Figure 4a. These experiments should be performed using virally-transduced intestinal organoids, which are much more robust and appropriate than NIH3T3 cells. This would also support or contrast the findings described by Neumeyer et al. 2018.

We agree that the recently established organoid model faithfully reflects biological characteristics of cancer. We also believe that the organoid model is an appropriate and strong tool for examining the requirement and dependency of growth factors essential for maintaining organoid growth, which can support our results in tumorigenesis experiments at least partially. Indeed, we already demonstrated the change of growth factor requirement with the use of organoid model. A RNF43-3SA dominant

negative phospho-mutation confers Rspo independency to organoid despite that all RNF43 mutants retain Wnt dependency (Figs. 3c, S5d and S5e), as described in our response to comment #4. Furthermore, we also showed that the constitutive activation of RNF43-3SD compromises the long-term maintenance of organoids even in the presence of enough Wnt ligand and Rspo (Figs. 3d and S5f). These results clearly show that the RNF43 phospho-status is an important regulation factor for the growth and maintenance of intestinal stem cells.

In addition, we argue on the suggested experimental setup. The intestinal organoid is a non-transformed primary cell with intact number of chromosomes and genetic stability. It is well known that at least four genes (*APC*, *KRAS*, *SMAD4*, *TP53*) have to be mutated to show proper engraftment and formation of cancer. The main purpose of our experiment in Figure 4 is to show the cooperation of active *Ras* and mutant *RNF43* in tumorigenesis. In theory, such cooperation would not be observed in organoid allograft model as there will not be any tumour to be formed. This phenomenon is reported by two independent groups in Nature and Nature Medicine (Drost et al. Nature 2015, DOI: 10.1038/nature14415, Matano et al. Nat Med 2015, DOI: 10.1038/nm.3802). Therefore, we believe that our choice of models (NIH3T3 and Cle-H3) was still relevant to establish the synergistic effect of active Ras and mutant RNF43.

6. Figure 4b-d. Given the authors highlight the co-occurrence of RNF43 and KRAS mutation in pancreatic cancer, pancreatic cancer cell lines mutant for Kras (AsPC1, DANG, Capan1) should be used instead of Cle-H3 cells.

Recently, it was reported that CRC cells established via a multistep tumorigenesis with mutations in *KRAS* /*BRAF*, *SMAD4* and *TP53*, which acquired the ability to grow independently from initial Wnt activation (Han et al. bioRxiv 2020, DOI: 10.1101/2020.01.22.914689). Our unpublished results in the intestine of *APC^{min};BAT-LacZ* Wnt reporter mice show that small adenomas display a strong signal of Wnt reporter, but late stage of adenomas or carcinomas lose the sign of Wnt activity (not shown here due to

the data for another research project), supporting the results in the above manuscript.

Similarly, the database analysis in COSMIC for suggested cell lines showed that all these *KRAS* mutant pancreatic tumour cells are carrying mutations in *KRAS/BRAF*, *SMAD4* and *TP53* (AsPC-1 cells, *BRAF*+ *KRAS*+ *RNF43*+ *SMAD4* + *TP53* + other 23 cancerous mutations; DAN-G cells, *KRAS* + *SMAD4* + *TP53* + other 30 cancerous mutations; Capan-1 cells, *KRAS* + *SMAD4* + *TP53* + *TCF7L2* + other 24 cancerous mutations). Therefore, we believe that it is extremely difficult to determine the appropriate model cell line that maintains Wnt dependency and suitable for our purpose. This point is so essential for us to accurately understand the data obtained from these cells.

Comparably, the traditional NIH3T3 cells are well-proven non-tumour cells, which do not form a tumour upon allograft transplantation, and Cle-H3 cells are a 'non-transformed *KRAS* mutant' model with a minimum allograft transformation activity (Figs. 4a and S7c). Therefore, we believe that these models are more appropriate than the suggested cell lines to elucidate the synergy between Ras and RNF43 in the onset of tumorigenesis.

7. Do the RNF43-3SA tumour xenografts display changes in WNT signalling/high nuclear β -catenin? This should be examined.

It is very well known that nuclear β -catenin is observed in the *APC* mutant mouse models. However, not all Wnt activation models result in a clear nuclear β -catenin expression. For example, we did not observe a strong nuclear localisation of β -catenin in *RNF43/ZNRF3* DKO intestinal tumours, although the expression of Wnt target genes strongly suggested high Wnt activity (Koo et al. Nature 2012, DOI: 10.1038/nature11308).

Nevertheless, we followed the reviewer's suggestion and performed immunohistochemical staining for nuclear β -catenin in RNF43-3SA allograft. We observed that a not so strong nuclear β -catenin but a relatively strong cytoplasmic accumulation of β -catenin in RNF43-3SA sample compared to that of mock Cle-H3 cells (Fig. S7c, shown below). This is very much in line

with our previous observation (Koo et al. Nature 2012, DOI: 10.1038/nature11308) where we also saw a relatively stronger cytoplasmic staining of β -catenin with strong activation of Wnt downstream target genes. We additionally presented this result (Fig. S7d) and added it to the main text (page 13, lane 22-24).

8. The link between p53 and phosphorylation of RNF43 is not convincing, and relies on the expression of p21 and Bax following DNA damage. WNT is sufficient to repress p21 via Myc in both p53 WT and mutant CRC cells (van de Wetering, 2002, Cell). Shouldn't you see some changes in p21 in your RNF43 mutant conditions, given they alter WNT signalling? If phosphorylation of RNF43 regulates aspects of p53 signalling, but not expression, would it be worthwhile testing RNF43 mutant constructs in p53-mutant CRC cells?

We appreciate the concern of the reviewer that p21 can be upregulated with an alternative pathway, e.g. Wnt-myc-p21 pathway instead of our proposed link, p53. In order to provide a clear evidence that RNF43 suppresses p21 and Bax via p53, we examined the expression of these p53 target genes in the repression of Wnt-myc activation in HCT-116 cells with iCRT3 or in the absence of p53 expression using p53 KO MEF-derived (MB352) cells. MB352 cells lost p21 expression completely due to the lack of p53, whereas etoposide treatment enhances p21 expression in NIH3T3 (Fig. S8b). Another

p53 target, Bax expression, is not perturbed by RNF43 and its mutants in the absence of p53 regardless of the presence or absence of etoposide (shown below).

Furthermore, inhibiting the binding of β -catenin to Tcf/Lef with iCRT3 in HCT116 cells suppressed Wnt signalling and downstream c-myc expression (Fig. S8a). At the same time, p21 expression is equally increasing with iCRT3 treatment regardless of RNF43 activity in Wnt signalling, suggesting that the Wnt-myc-p21 pathway functions in HCT116 cells. However, RNF43 and its mutants are not involved in the Wnt-myc-p21 pathway because changes of RNF43 function do not result in the down-/upregulation of active β -catenin, c-myc and Wnt signalling in the absence of Wnt ligand (Fig. S8a, shown below) as shown in Fig. S1h and described in our response to comment #4.

This additional data is consistent with our previous reports which show the inhibitory role of RNF43 in p53 pathway (page 14, line 6–12) and supports our finding that RNF43 and its mutants similarly suppress p53 function regardless of their differential effect to the Wnt activity.

9. With respect to co-operation between Ras and RNF43 in CRC, a more appropriate mutation combination is Braf and RNF43. Perhaps, the authors could test the efficacy of their RNF43 mutant constructs in organoids from Braf-mutant mice. This approach would better reflect the serrated CRC subtype.

As the reviewer pointed out, it is reported that G659fs mutation in *RNF43* is frequently identified in MSI-CRC and this mutation is highly correlated with the *BRAF* mutation (Bond et al. Oncotarget 2016, DOI: 10.18632/oncotarget.12130). However, the combination of these mutations is mostly found in tumours with MSI phenotype (Li et al. Oncogene 2020, DOI:10.1038/s41388-020-1232-5). It is known that G659fs mutation is a hotspot of mutagenesis in MSI cancers. But several reports suggested that G659fs retains its activity and equally suppresses Wnt signalling which is the same as wild-type RNF43 when overexpressed in cells (Tu et al. Sci Rep 2019, DOI: 10.1038/s41598-019-54931-3, Li et al. Oncogene 2020, DOI: 10.1038/s41388-020-1232-5). However, it is not clear whether G659fs still allows the translation of functional RNF43 protein to suppress Wnt signalling or does not expressed due to nonsense-mediated mRNA decay (NMD) caused by premature stop codon. It is also well known that MSI cancer patients with *RNF43^{G659fs}* and *BRAF* mutations frequently show better prognosis than MSS patients. Altogether, we still do not know the exact role of *RNF43^{G659fs}* mutation in the tumorigenesis with MSI phenotype. Nevertheless, the main focus of our manuscript is the role of phospho-switch of RNF43 in the progression of multistep tumorigenesis and not to understand all known RNF43 mutants including G659fs. Therefore, we believe that it is not essential in this manuscript to assess the linked roles of *BRAF* and *RNF43^{G659fs}* in serrated CRC subtype but should be discussed in another research project in the future.

Additional point

As the authors show that the RNF43-HA tag acts in a similar manner to endogenous RNF43. I am comfortable with the use of this reagent, endogenous RNF43 is not so easy to work with.

We are very pleased to see that the reviewer #3 agrees with our answers to all the concerns raised repeatedly by Reviewer #1 that exogenously introduced RNF43 and its mutants function similarly to endogenous ones as shown in Fig. S3.

REVIEWERS' COMMENTS:

Reviewer #3 (Remarks to the Author):

The revised manuscript from Tsukiyama et al. has addressed all of the concerns and suggestions raised via additional experiments and clearer explanations in the main text. This version is now acceptable for publication.

Point-by-point response to Reviewers' comments (shown in blue):

Reviewer #3 (Remarks to the Author):

The revised manuscript from Tsukiyama et al. has addressed all of the concerns and suggestions raised via additional experiments and clearer explanations in the main text.

This version is now acceptable for publication.

RESPONSE: We thank the reviewer for this comment.